# SUCLG1 restricts POLRMT succinylation to enhance mitochondrial biogenesis and leukemia progression

Weiwei Yan [1,10], Chengmei Xie[1,10], Sijun Sun[1,2,10], Quan Zheng [3,10], Jingyi Wang[4], Zihao Wang [5], Cheuk-Him Man[6], Haiyan Wang[4], Yunfan Yang [7], Tianshi Wang [8], Leilei Shi[1], Shengjie Zhang [1✉], Chen Huang [2✉], Shuangnian Xu [9✉] & Yi-Ping Wang [1,5✉]

## Abstract

**Mitochondria are cellular powerhouses that generate energy through the electron transport chain (ETC). The mitochondrial genome (mtDNA) encodes essential ETC proteins in a compartmentalized manner, however, the mechanism underlying metabolic regulation of mtDNA function remains unknown. Here, we report that expression of tricarboxylic acid cycle enzyme *succinate-CoA ligase SUCLG1* strongly correlates with ETC genes across various TCGA cancer transcriptomes. Mechanistically, SUCLG1 restricts succinyl-CoA levels to suppress the succinylation of mitochondrial RNA polymerase (POLRMT). Lysine 622 succinylation disrupts the interaction of POLRMT with mtDNA and mitochondrial transcription factors. SUCLG1-mediated POLRMT hyposuccinylation maintains mtDNA transcription, mitochondrial biogenesis, and leukemia cell proliferation. Specifically, leukemia-promoting *FMS-like tyrosine kinase 3* (*FLT3*) mutations modulate nuclear transcription and upregulate *SUCLG1* expression to reduce succinyl-CoA and POLRMT succinylation, resulting in enhanced mitobiogenesis. In line, genetic depletion of *POLRMT* or *SUCLG1* significantly delays disease progression in mouse and humanized leukemia models. Importantly, succinyl-CoA level and POLRMT succinylation are downregulated in *FLT3*-mutated clinical leukemia samples, linking enhanced mitobiogenesis to cancer progression. Together, SUCLG1 connects succinyl-CoA with POLRMT succinylation to modulate mitochondrial function and cancer development.**

**Keywords** FMS-like Tyrosine Kinase 3; Lysine Succinylation; Mitochondrial Biogenesis; Mitochondrial RNA Polymerase; Succinate-CoA Ligase
**Subject Categories** Cancer; Haematology; Metabolism

## Introduction

Mitochondria are membrane-bound organelles that serve as powerhouses in eukaryotes (Nunnari and Suomalainen, 2012; Roger et al, 2017). Distinct from other organelles, mitochondria harbor unique genomes (mtDNA) to encode essential components in the electron transport chain (ETC) (Kopinski et al, 2021). mtDNA resides in the mitochondria matrix where organelle-specific transcription complex mediates mRNA synthesis and gene expression (Gustafsson et al, 2016). As organelles that are highly specialized in energy production, mitochondria tightly couple mtDNA transcription with cellular metabolic status (Liu and Birsoy, 2023). However, the mechanism underlying the metabolic regulation of mtDNA function remains unknown.

The mitochondrial transcription complex comprises mitochondrial RNA polymerase (POLRMT) and mitochondrial transcription factors A and B2 (TFAM and TFB2M) (Scarpulla, 2008). TFAM and TFB2M facilitate the recognition of transcription start sites and mtDNA binding of POLRMT, allowing it to mediate the transcription of mtDNA-encoded genes (Kravchenko et al, 2005). POLRMT is a key regulator of both mtDNA replication and transcription as it also synthesizes the RNA primer necessary for mitochondrial genome duplication (Kuhl et al, 2016). Previous studies suggest that the activity of mitochondrial transcription complex is intimately linked to cell metabolism: (1) mitochondrial gene expression is dynamically connected to nutrient supply and bioenergetic requirements (Liesa and Shirihai, 2013; Scarpulla, 2008); (2) genetic mutations of various metabolic enzymes, such as succinate-CoA ligase subunit alpha (SUCLG1), succinate-CoA ligase [ADP-forming] subunit beta (SUCLA2), thymidine kinase 2 (TK2), and deoxyguanosine kinase (DGUOK), are clinically associated with mtDNA depletion syndrome (MDS) (Copeland, 2008). Of note, TK2 and DGUOK are well-established regulators of nucleotide metabolism and provide precursors for mtDNA synthesis (Mandel et al, 2001; Saada et al, 2001). In contrast, the regulatory role of SUCLG1 and SUCLA2, which are TCA cycle

---

[1]Precision Research Center for Refractory Diseases, Institute for Clinical Research, Shanghai Key Laboratory of Pancreatic Disease, Shanghai General Hospital, Shanghai Jiao Tong University School of Medicine, 200080 Shanghai, China. [2]Department of Gastrointestinal Surgery, Shanghai General Hospital, Shanghai Jiao Tong University School of Medicine, 200080 Shanghai, China. [3]Center for Single-Cell Omics, School of Public Health, Shanghai Jiao Tong University School of Medicine, Shanghai 200025, China. [4]Department of Ophthalmology, Shanghai General Hospital, Shanghai Jiao Tong University School of Medicine, 200080 Shanghai, China. [5]Institutes of Biomedical Sciences, Shanghai Medical College, Fudan University, 200032 Shanghai, China. [6]Division of Haematology, Department of Medicine, Li Ka Shing Faculty of Medicine, The University of Hong Kong, Hong Kong SAR, China. [7]Department of Cell Biology, School of Basic Medical Sciences, Cheeloo College of Medicine, Shandong University, 250012 Jinan, China. [8]Department of Biochemistry and Molecular Cell Biology, Shanghai Jiao Tong University School of Medicine, 200025 Shanghai, China. [9]Department of Hematology, Southwest Hospital, Army Medical University, 400038 Chongqing, China. [10]These authors contributed equally: Weiwei Yan, Chengmei Xie, Sijun Sun, Quan Zheng. ✉E-mail: shengjie.zhang@shgh.cn; huangchen0204@sjtu.edu.cn; xushuangnian@tmmu.edu.cn; yiping.wang1@shgh.cn

enzymes, in mtDNA replication and transcription remains unclear (Ostergaard et al, 2007). We therefore speculate that succinate-CoA ligases potentially connect cellular metabolic status to mtDNA.

Mitochondria produce a variety of metabolic intermediates that may act as signaling molecules to modulate mitochondrial biogenesis (mitobiogenesis) (Wang et al, 2021). These intermediate metabolites could also serve as precursors for post-translational modifications (PTM) (Diskin et al, 2021). Proteomic studies have demonstrated that mitochondrial replication and transcription complexes are modified in forms of phosphorylation, acetylation, and succinylation (King et al, 2018; Reardon and Mishanina, 2022; Wang et al, 2021). Notably, succinylation is a PTM during which a succinyl group is conjugated to a lysine residue. Succinyl-CoA provides the succinyl group and chemically modifies the target protein (Zhang et al, 2011). Previously, succinylation of metabolic enzymes has been shown to modulate energy metabolism and redox balance in different cancers (Li et al, 2015; Tong et al, 2021; Zhou et al, 2016). The regulatory role of protein succinylation in mitobiogenesis remains unclear.

Mitobiogenesis maintains energy production to support cell proliferation and cancer development (Vyas et al, 2016). Oncogenic mutations remodel mitobiogenesis programs to promote cancer initiation and progression. We performed pan-cancer correlation analysis of TCGA transcriptomes and found that, among mitochondrial metabolic genes, *SUCLG1* strongly showed a positive relation with ETC gene expression. Mechanistically, SUCLG1 restricted POLRMT succinylation to enhance mtDNA transcription, connecting succinyl-CoA signal to mitobiogenesis. Acute myeloid leukemia (AML), which is dependent on mitochondrial metabolism (Chen et al, 2019; Ishizawa et al, 2019), is driven by a series of genomic mutations arised in the bone marrow cells (Li et al, 2016). AML-derived mutations of *FMS-like tyrosine kinase 3* (*FLT3*) upregulated SUCLG1 to induce POLRMT hyposuccinylation, resulting in enhanced mitobiogenesis and leukemia progression.

# Results

## SUCLG1 maintains mtDNA-encoded gene expression and mitochondrial mass

To identify which metabolic enzymes potentially regulate mtDNA function, we analyzed the correlation between mitochondrial metabolic genes and ETC genes in TCGA transcriptomic studies (Fig. 1A). Among 405 mitochondrial genes involved in metabolism according to MitoCarta3.0 (Rath et al, 2021), 397 genes have valid Spearman's correlation with ETC genes across 32 different cancer types in TCGA Firehose legacy datasets (Gao et al, 2019). Pan-cancer ranking of ETC correlation coefficients revealed that *SUCLG1* was strongly associated with ETC gene expression across various cancers (Fig. 1A,B). SUCLG1 converts succinyl-CoA to succinate, both metabolites are important signaling molecules in cell metabolism (Fig. EV1A) (Gu et al, 2021; Wu et al, 2020b). We adopted AML as a model as mitochondrial respiration is indispensable to leukemic growth (Farge et al, 2017). Gene set enrichment analysis revealed a positive correlation of *SUCLG1* with ETC, also known as oxidative phosphorylation, in three independent AML transcriptomic studies (Fig. EV1B) (Data ref: Saksena

et al, 2023; Data ref: Sabatier et al, 2023a; Data ref: Naldini et al, 2023). Ranking of mitochondrial enzymes in the TCGA AML dataset demonstrated that *DGUOK*, which provides building blocks for mtDNA synthesis (Mandel et al, 2001), showed the most prominent ETC correlation (Fig. EV1C). This observation supports the robustness of our correlation analysis. We next asked whether SUCLG1 acted as a positive regulator of mitobiogenesis. To test this, we stably expressed two different short-hairpin RNAs (shRNAs) against *SUCLG1* in human leukemia cells. CD34[+] cord blood (CB) cells were included as the normal control (Figs. 1C and EV1D). MitoTracker Green (MTG) is a fluorescent dye that specifically labels mitochondria. As anticipated, depleting *SUCLG1* remarkably reduced the mtDNA abundance and MTG intensity of tested cells (Figs. 1C and EV1D). Besides, loss of *SUCLG1* decreased mitochondrial mass in cell lines derived from colorectal cancer (HCT116), lung cancer (A549), and liver cancer (HepG2) (Figs. 1C and EV1D). Collectively, SUCLG1 positively regulates mitochondrial mass.

Because SUCLG1 locates in the mitochondrial matrix, we hypothesized that SUCLG1 might regulate organelle-specific mitobiogenesis. Supporting this idea, depletion of *SUCLG1* in AML cell lines (HL60 and MV411) decreased the mRNA and protein expression of mtDNA-encoded genes (Fig. 1D,E). The expression of DNA polymerase subunit gamma (POLG), POLRMT, and mitochondrial ribosomal protein L45 (MRPL45), which are key regulators of mtDNA replication, transcription, and protein translation (Rahman and Copeland, 2019; Scarpulla, 2008; Wang et al, 2021), were minimally altered (Fig. 1D,E). In addition, nuclear factors governing mitochondrial biogenesis, including peroxisome proliferator-activated receptor-γ coactivator-1α (PGC-1-α) and nuclear factor erythroid 2-related factor 2 (NRF2) (Scarpulla, 2008) showed mild changes (Fig. 1D,E). These results imply that SUCLG1 maintains mtDNA-encoded gene expression. To validate the role of SUCLG1 in mitobiogenesis, we performed electron microscopy analysis to quantify mitochondria in MV411 cells. In agreement with previous results, mitochondria number in *SUCLG1*-knockdown cells was reduced by ~50% (Fig. 1F,G). We further isolated mitochondria from stable cells and assayed the activities of ETC complexes (Fig. EV1E). Depletion of *SUCLG1* significantly downregulated the catalytic activities of five different complexes (Fig. 1H). These data collectively support that SUCLG1 maintains mitobiogenesis.

We next tested the metabolic role of SUCLG1 in leukemia cells. *SUCLG1* showed a positive correlation with other ETC-correlated metabolic genes in the majority of TCGA cancer types (Fig. EV1F). Two of the top-ranking ETC-related genes, *PRDX5* and *TXN2*, showed decreased mRNA expression after knocking down *SUCLG1* in both MV411 and HL60 cells (Fig. EV1G,H). These data indicate that ETC-correlated genes may work cooperatively to control mitochondrial activity. Next, we focused on SUCLG1 for detailed metabolic analysis. CellROX Green staining of MV411 cells revealed increased ROS levels after depleting *SUCLG1* (Fig. EV1I,J). In accordance, elevated apoptosis was observed in *SUCLG1*-knockdown cells in Annexin V/7-amino-actinomycin (7-AAD) staining assay (Fig. EV1K,L). Further, we treated MV411 cells with a low dose of doxorubicin to induce senescence as previously described (Victorelli et al, 2023). Loss of *SUCLG1* did not result in obvious changes in β-galactosidase staining (Fig. EV1M). Bromo-deoxyuridine (BrdU) incorporation was remarkably decreased in

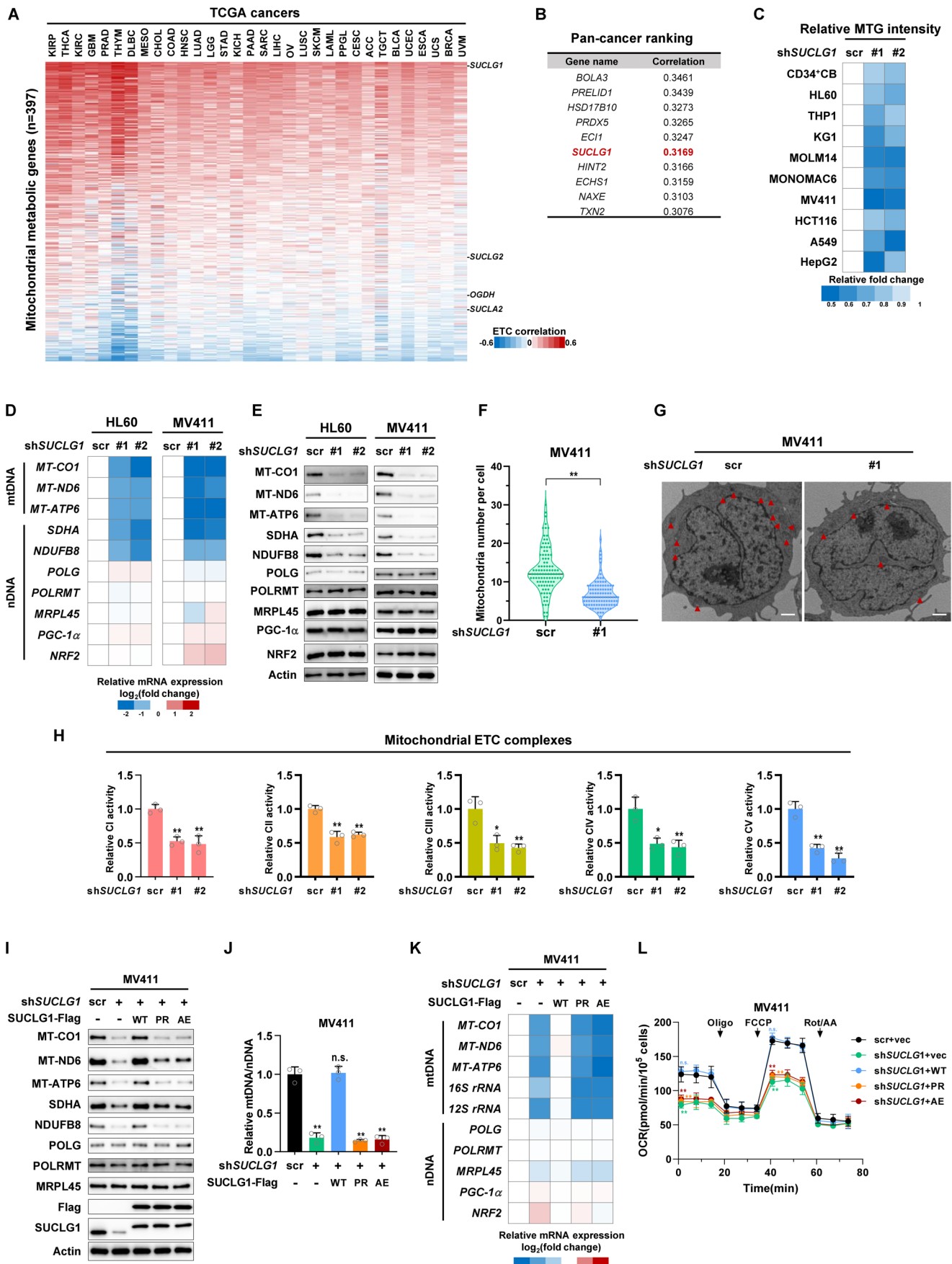

**Figure 1.  SUCLG1 maintains mtDNA-encoded gene expression and mitochondrial mass.**

(A) Spearman's correlation of mitochondrial metabolic genes with ETC genes was analyzed in various TCGA transcriptomes. (B) List of top ten mitochondrial metabolic genes that positively correlate with ETC genes expression in pan-cancer analysis. Shown are Spearman's correlation coefficients. (C) Scrambled control or shRNAs against *SUCLG1* were transduced into human CD34$^+$ cord blood (CB) cells or cancer cell lines as indicated. Mitochondrial mass was quantified by MTG staining. Shown is the relative MTG fluorescence intensity compared to scrambled control group. $n = 3$ independent biological replicates. (D, E) Total RNA was extracted from control or stable *SUCLG1*-knockdown HL60 and MV411 cells. Relative mRNA expression of mtDNA and nDNA-encoded genes was quantified by qPCR and displayed on a log$_2$ scale (D). $n = 3$ independent biological replicates. Protein expression was determined by western blotting (E). (F, G) Control and *SUCLG1*-knockdown MV411 cells were analyzed with transmission electron microscopy. Mitochondria numbers in 100 cells were determined (F). Shown were representative images (G), arrowheads indicate mitochondria. Scale bars: 1 μm. Data represent means ± SD, $n = 100$ different cells, $t$ test. **$P < 0.01$. (H) Mitochondria isolated from control or *SUCLG1*-knockdown MV411 cells were subjected to ETC complex activity assay. Complex activities were normalized to total mitochondrial protein. Data represent means ± SD, $n = 3$ independent biological replicates, $t$ test. **$P < 0.01$; *$P < 0.05$. (I–L) Wild-type SUCLG1 or its P170R (PR) and A209E (AE) mutants were re-expressed in *SUCLG1*-knockdown MV411 cells. Protein expression of mtDNA and nDNA-encoded genes was determined by western blotting (I). mtDNA abundance (J), mRNA expression levels (K), and oxygen consumption rates (L) of stable cells were determined. Data represent means ± SD, $n = 3$ independent biological replicates, $t$ test. **$P < 0.01$; n.s. not significant. Source data are available online for this figure.

*SUCLG1*-knockdown cells, indicating reduced cell proliferation (Fig. EV1N). Supporting this observation, cell cycle analysis demonstrated that *SUCLG1*-depleted cells were less actively proliferating compared to the scrambled control (Fig. EV1O). Together, SUCLG1 supports the proliferation of leukemia cells.

Human inborn *SUCLG1* mutations have been clinically linked to mtDNA depletion syndrome with an unknown mechanism (Ostergaard et al, 2007). Disease-derived mutations target proline 170 (P170) and alanine 209 (A209), which are highly conserved residues during evolution and reside in close proximity to the catalytic site of SUCLG1 (Huang and Fraser, 2021) (Fig. EV1P–R). We constructed Flag-tagged disease-derived SUCLG1 mutants, P170R and A209E (Chinopoulos et al, 2019; Rouzier et al, 2010), to test their biological impacts. Enzymatic activity assay using immunopurified SUCLG1-Flag protein demonstrated that P170R and A209E mutants exhibited deficient catalysis compared to the wild-type enzyme (Fig. EV1S). Further, we re-expressed SUCLG1$^{P170R}$ and SUCLG1$^{A209E}$ into *SUCLG1*-knockdown MV411 cells (Fig. 1I). SUCLG1 mutants failed to restore mtDNA abundance, mtDNA-encoded gene expression, and cellular respiration (Fig. 1I–L). These data suggest that SUCLG1 activity is essential for maintaining mitochondrial mass and respiration.

## SUCLG1 reduces succinyl-CoA level to restrict POLRMT succinylation

We next asked how SUCLG1 modulated organelle-specific biogenesis. Succinate-CoA ligase is a heterodimeric enzyme (Wu et al, 2020a). SUCLG1 associates with SUCLG2 or SUCLA2 to convert succinyl-CoA to succinate and simultaneously generate GTP or ATP, respectively (Fig. 2A). Succinyl-CoA is derived from α-ketoglutarate (α-KG) by oxoglutarate dehydrogenase (OGDH) (Fig. 2A). Unlike *SUCLG1*, neither *SUCLG2*, *SUCLA2* nor *OGDH* showed a consistent correlation with ETC gene expression in TCGA transcriptomes (Fig. EV2A–C). Protein expression of OGDH remained unchanged in *SUCLG1*-knockdown cells (Fig. EV2D). Therefore, we focused on SUCLG1 to study how metabolism linked to mitobiogenesis. We speculate that SUCLG1 might alter mitobiogenesis through controlling the abundance of specific metabolites. Metabolome profiling demonstrated that glycolytic intermediates were increased in *SUCLG1*-knockdown MV411 cells (Fig. 2B). Amino acids were also dysregulated after depleting *SUCLG1*. Strikingly, succinyl-CoA (SucCoA) was accumulated in *SUCLG1*-deficient cells (Fig. 2B). We next asked how

cancer cells survived the mitochondrial defect in the absence of *SUCLG1*. Metabolic flux assay showed that *SUCLG1*-knockdown cells had increased basal glycolytic rate (Figs. 2C and EV2E), indicating that glycolysis may be enhanced to compensate for mitochondria respiration defect. Supporting this idea, glucose consumption and lactate production were elevated after *SUCLG1* depletion in both MV411 and HL60 cells (Figs. 2D,E and EV2F,G). Quantification of cellular ATP validated that energy supply was reduced in *SUCLG1*-depleted MV411 and HL60 cells (Figs. 2F and EV2H). In contrast to glucose uptake, glutamine consumption was mildly decreased in *SUCLG1*-depleted MV411 cells (Fig. 2G). Therefore, glycolysis is upregulated in *SUCLG1*-deficient cells to cope with mitochondria dysfunction and maintain cell survival.

We next monitored metabolic changes in *SUCLG1*-knockdown and rescue cells. In *SUCLG1*-knockdown cells, elevated α-KG and succinyl-CoA were accompanied by reduced ATP and GTP levels (Fig. 2H). Re-expression of wild-type SUCLG1, but not its catalysis-deficient mutants, restored succinyl-CoA abundance (Fig. 2H). Of note, succinate was not dramatically decreased. We tested the mRNA expression of genes related to succinate metabolism and transport (Fig. EV2I). SLC13A5, a cell membrane-bound transporter, showed increased expression after knocking down *SUCLG1* (Fig. 2I). Other succinate transporters only demonstrated modest changes in mRNA expression. Hence, SLC13A5 may import exogenous succinate to compensate for decreased succinate production in *SUCLG1*-knockdown cells. Succinyl-CoA exists in a compartmentalized manner in mitochondria (Iovine et al, 2021). We further isolated mitochondria to quantify succinyl-CoA, which was altered in a similar pattern with cellular succinyl-CoA (Figs. 2J and EV2J). Therefore, SUCLG1 restrains mitochondrial succinyl-CoA, which may contribute to its role in regulating mitobiogenesis.

Succinyl-CoA serves as a succinyl donor to modify substrate protein in succinylation reactions. We speculated that SUCLG1 might modify protein succinylation. The distribution of metabolites is highly heterogeneous within cells (Li et al, 2023). Metabolic enzymes have been demonstrated to alter metabolite levels and efficiently modulate the activity of their interacting proteins (Huang et al, 2023; Wang et al, 2019). To identify potential targets of SUCLG1, we assayed the interaction between SUCLG1 and regulators of mtDNA replication and transcription. Intriguingly, endogenous SUCLG1 readily associated with POLRMT and TFAM, but not POLG, in MV411 cells (Fig. 2K). We focused on POLRMT as it directly mediates mtDNA transcription. We immunopurified

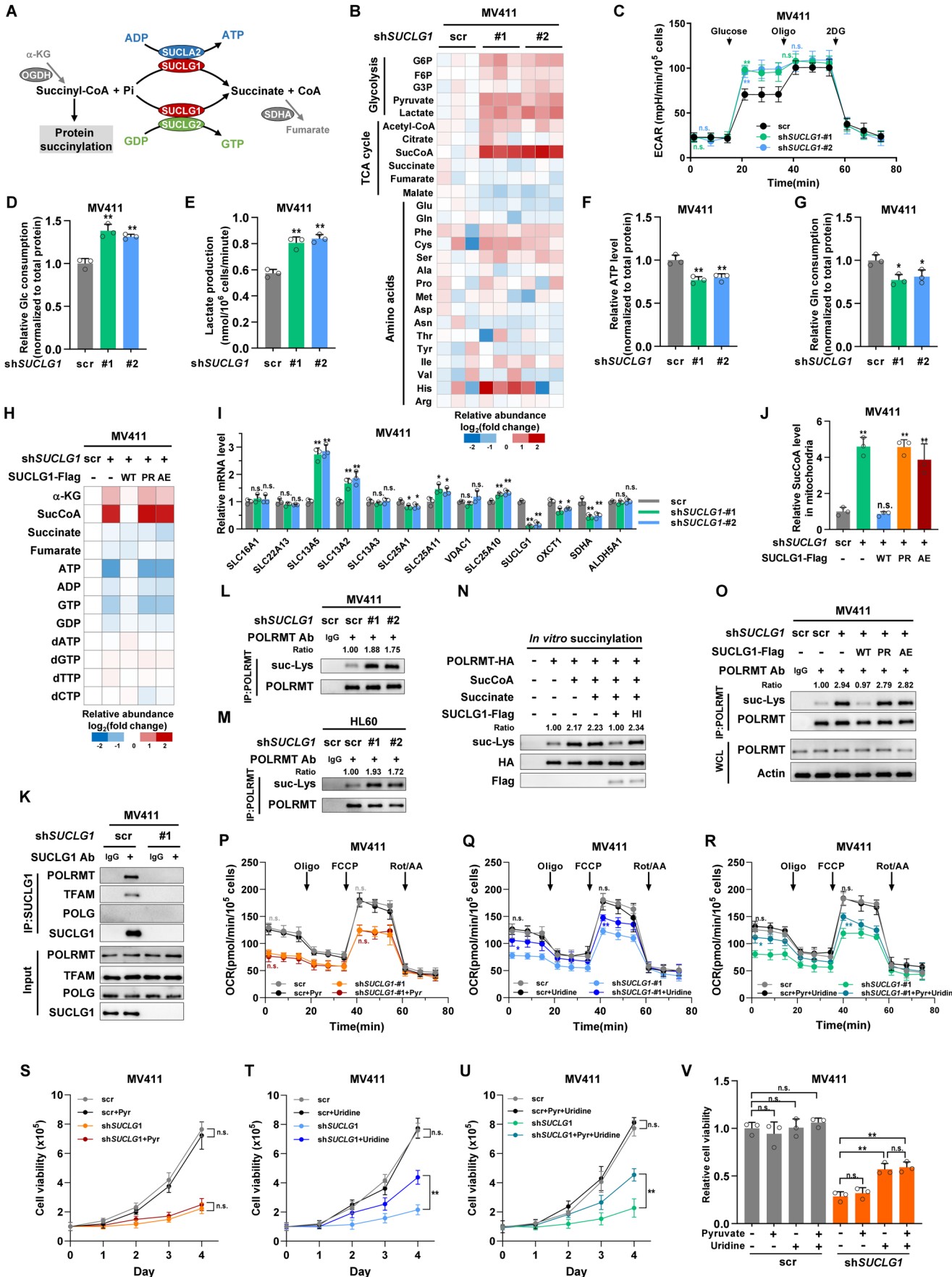

◄  **Figure 2.   SUCLG1 reduces succinyl-CoA level to restrict POLRMT succinylation.**

(A) Schematic overview of reactions catalyzed by succinyl-CoA ligase complex. SDHA succinate dehydrogenase subunit A, OGDH oxoglutarate dehydrogenase, SUCLG2 succinate-CoA ligase [GDP-forming] subunit beta, Pi phosphate. (B) Metabolites were extracted from control or *SUCLG1*-knockdown MV411 cells, and quantified by mass spectrometry. The relative abundance of indicated metabolites was shown on a $\log_2$ scale. $n = 3$ independent biological replicates. (C) Extracellular acidification rates of control or *SUCLG1*-knockdown MV411 cells were determined. Data represent means ± SD, $n = 3$ independent biological replicates, $t$ test. **$P < 0.01$; n.s. not significant. (D, E) Glucose (Glc) consumption and lactate production were assayed in control and *SUCLG1*-knockdown MV411 cells. Data represent means ± SD, $n = 3$ independent biological replicates, $t$ test. **$P < 0.01$. (F) ATP abundance was quantified in control and *SUCLG1*-knockdown MV411 cells, and normalized to total protein. Data represent means ± SD, $n = 3$ independent biological replicates, $t$ test. **$P < 0.01$. (G) Glutamine (Gln) consumption was assayed in control and *SUCLG1*-knockdown MV411 cells. Data represent means ± SD, $n = 3$ independent biological replicates, $t$ test. *$P < 0.05$. (H) Metabolites were extracted from *SUCLG1*-knockdown and rescue MV411 cells, and quantified by mass spectrometry. The relative abundance of indicated metabolites was shown on a $\log_2$ scale. $n = 3$ independent biological replicates. (I) Total RNA was extracted to determine the mRNA expression of succinate-related metabolic genes in control and *SUCLG1*-knockdown cells. Data represent means ± SD, $n = 3$ independent biological replicates, $t$ test. **$P < 0.01$; *$P < 0.05$; n.s. not significant. (J) Mitochondria were isolated from stable MV411 cells. Metabolites were extracted to determine succinyl-CoA abundance. Data represent means ± SD, $n = 3$ independent biological replicates, $t$ test. **$P < 0.01$; n.s. not significant. (K) Endogenous SUCLG1 was immunoprecipitated from MV411 cells. The protein interaction of SUCLG1 was determined by western blotting. IgG was included as a negative control for immunoprecipitation. *SUCLG1*-knockdown cells were included to validate the specificity of SUCLG1 antibody. (L, M) Endogenous POLRMT was immunoprecipitated from control or *SUCLG1*-knockdown MV411 (L) and HL60 (M) cells. Lysine succinylation was determined by western blotting. IgG was included as a negative control for immunoprecipitation. (N) Immunopurified POLRMT-HA protein was incubated with succinyl-CoA (sucCoA) or succinate. Flag-tagged SUCLG1 protein was immunopurified and treated with or without heat-inactivation (95 °C for 10 min). POLRMT succinylation was determined after in vitro succinylation. (O) Endogenous POLRMT was immunoprecipitated from *SUCLG1*-knockdown and rescue MV411 cells. Lysine succinylation was determined by western blotting. IgG was included as a negative control for immunoprecipitation. (P–R) Scrambled control or *SUCLG1*-knockdown (sh-#1) MV411 cells were cultured with 1 mM pyruvate or 0.2 mM uridine for 24 h. Oxygen consumption rates were determined. Data represent means ± SD, $n = 3$ independent biological replicates, $t$ test. **$P < 0.01$; *$P < 0.05$; n.s. not significant. (S–V) Scrambled control or *SUCLG1*-knockdown (sh-#1) MV411 cells were cultured with 1 mM pyruvate or 0.2 mM uridine. Cell proliferation was determined. Data represent means ± SD, $n = 3$ independent biological replicates, $t$ test. **$P < 0.01$; n.s. not significant. Source data are available online for this figure.

HA-tagged POLRMT from control and *SUCLG1*-knockdown cells to test its post-translation modifications. Loss of *SUCLG1* elevated POLRMT lysine succinylation (suc-Lys), but not its lysine methylation (me-Lys), lysine acetylation (ac-Lys), or tyrosine phosphorylation (p-Tyr) (Fig. EV2K). Besides, the ubiquitination of ectopically expressed or endogenous POLRMT didn't show remarkable changes (Fig. EV2L,M). Of note, succinylation levels of endogenous POLRMT were also upregulated in *SUCLG1*-knockdown MV411 and HL60 cells (Fig. 2L,M). These results demonstrate that SUCLG1 restricts POLRMT succinylation.

Succinylation occurs in a non-enzymatic fashion, allowing a direct linkage between mitochondrial succinyl-CoA levels and target protein succinylation (Gut et al, 2020; Wagner et al, 2017). To interrogate whether succinyl-CoA acts as a metabolic signal to modify POLRMT, we purified bacterially expressed His-tagged POLRMT (POLRMT-His) protein and carried out in vitro succinylation assay. Incubation with succinyl-CoA, but not succinate, dose-dependently increased POLRMT succinylation (Fig. EV2N). Next, we immunopurified HA-tagged POLRMT and Flag-tagged SUCLG1 from MV411 cells, and pre-incubated the reaction buffer with SUCLG1-Flag protein. In vitro succinylation reaction revealed that SUCLG1-Flag enzyme, but not its heat-inactivated form, suppressed POLRMT succinylation (Fig. 2N). Together, SUCLG1 restricted succinyl-CoA-induced POLRMT succinylation in vitro. Further, we immunopurified POLRMT from MV411 stable cells. Re-expressing wild-type SUCLG1, but not its mutants, restored lysine succinylation of endogenous POLRMT (Fig. 2O). Taken together, SUCLG1 restricts succinyl-CoA level to downregulate POLRMT succinylation.

Mitochondrial transcription is dependent on a sufficient supply of energy and nucleotides (Basu et al, 2020). To test the functional relevance between SUCLG1 and POLRMT, we treated *SUCLG1*-deficient MV411 cells with pyruvate and uridine, which support mitochondrial respiration and RNA synthesis, respectively. Supplementing with uridine, but not pyruvate, partially rescued the defect of mitochondrial respiration and cellular proliferation of *SUCLG1*-depleted cells (Figs. 2P–V and EV2O). Combined

treatment with both pyruvate and uridine didn't further increase oxygen consumption and cell growth (Figs. 2P–V and EV2O). These observations suggest that *SUCLG1*-deficient cells are auxotrophic for uridine. SUCLG1 is potentially involved in regulating mitochondrial transcription.

## K622 succinylation suppresses POLRMT activity

We next investigated the biochemical effect of POLRMT succinylation. Previous proteomic profiling indicated that lysine 622 (K622) of POLRMT, an evolutionarily conserved residue (Fig. EV3A), was potentially succinylated (Weinert et al, 2013). K622 locates at the DNA-binding groove of POLRMT (Schwinghammer et al, 2013) (Fig. 3A). Succinylation at K622 may impair mtDNA binding due to steric hindrance and affect the activity of mtDNA transcription. To precisely detect POLRMT succinylation, we generated a site-specific antibody against succinylated K622 (anti-K622sc) (Fig. EV3B). We also mutated K622 into arginine (K622R) or glutamate (K622E) to mimic its unmodified and succinylated states, respectively (Qi et al, 2019). We immunopurified HA-tagged POLRMT and its mutants. The anti-K622sc antibody specifically recognized wild-type POLRMT, but not K622R or K622E mutants (Fig. EV3C). Sirtuin 5 (SIRT5) is a mitochondrial deacylase responsible for removing succinylation modifications (Greene et al, 2019). Chemical inhibition of SIRT5 significantly increased K622 succinylation of wild-type POLRMT, but not its mutants (Fig. EV3C). These results demonstrate that K622 is a major succinylation site of POLRMT.

To determine the physiological function of POLRMT succinylation, we re-expressed HA-tagged POLRMT and its mutants into *POLRMT*-knockdown MV411 and HL60 cells (Fig. EV3D,E). Co-immunoprecipitation assay revealed that, compared to wild-type POLRMT, POLRMT^K622R showed stronger interaction with TFAM and TFB2M. In contrast, succinylation-mimicking POLRMT^K622E mutant showed deficient binding with mitochondrial transcription factors (Figs. 3B and EV3F). Further, we transduced a shRNA against *SUCLG1* into *POLRMT*-knockdown and rescue cells.

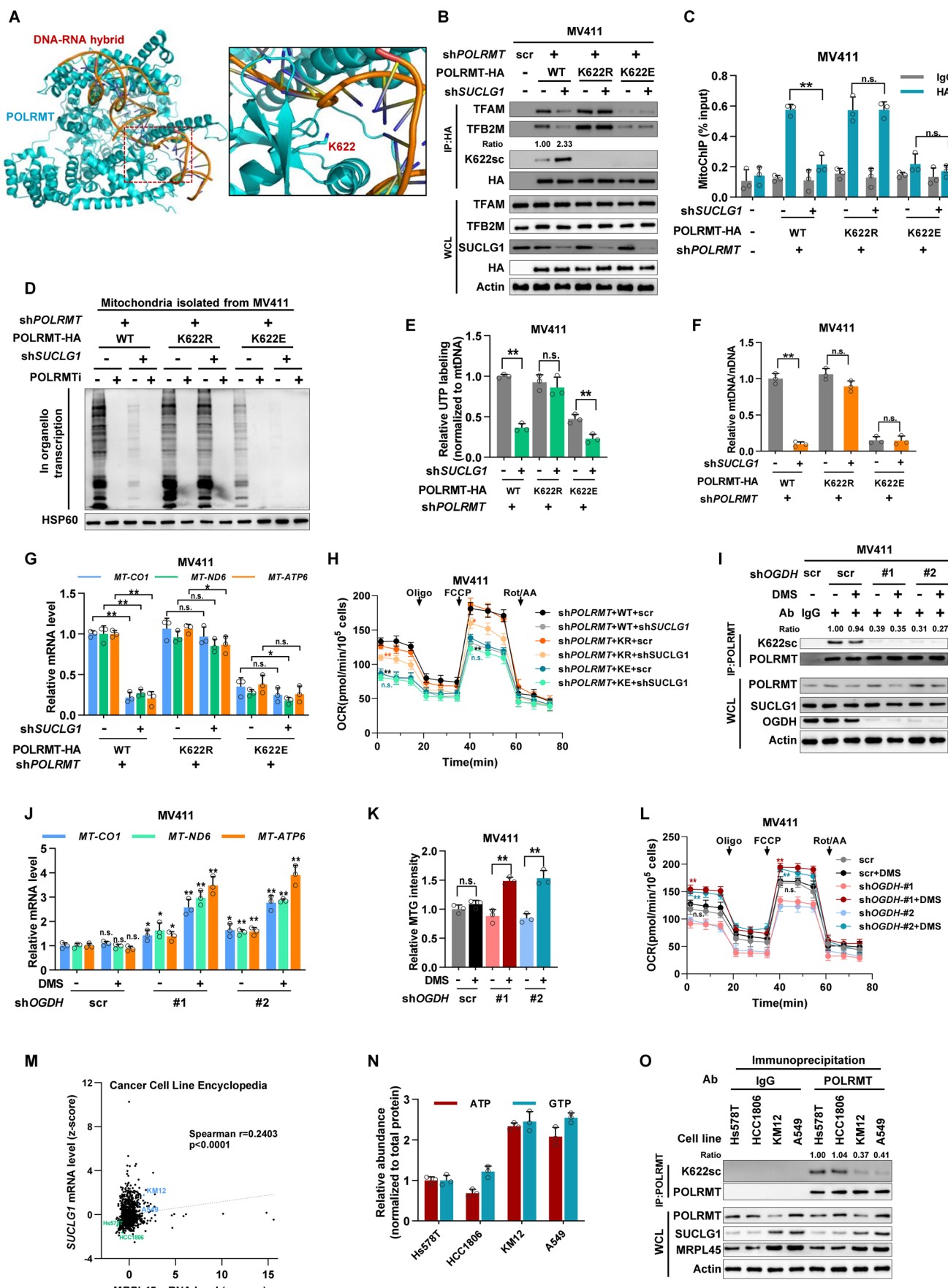

◄ **Figure 3. K622 succinylation suppresses POLRMT activity.**

(A) Structural illustration of POLRMT. Cyan indicates POLRMT protein, and the sugar-phosphate backbone of DNA-RNA hybrid was colored brown (PDB ID:4BOC). (B–E) HA-tagged POLRMT and its mutants were re-expressed in *POLRMT*-knockdown MV411 cells. Cells were further transduced with scrambled control or shRNA against *SUCLG1*. POLRMT-HA was immunopurified to determine its succinylation and protein interaction (B). MitoChIP was performed to evaluate mtDNA binding of POLRMT-HA (C). Mitochondria were isolated to evaluate in organello labeling of nascent RNA (D). UTP labeling was quantified and normalized to mtDNA (E). Data represent means ± SD, $n = 3$ independent biological replicates, one-way ANOVA with Dunnett's multiple comparisons test. **$P < 0.01$; n.s. not significant. (F–H) Relative mtDNA abundance (F), mRNA expression levels of mtDNA-encoded genes (G), and oxygen consumption rates (H) were determined in stable MV411 cells. Data represent means ± SD, $n = 3$ independent biological replicates, $t$ test. **$P < 0.01$; *$P < 0.05$; n.s. not significant. (I–L) Scrambled control or *OGDH*-knockdown stable MV411 cells were cultured with or without 50 μM dimethyl succinate (DMS). K622 succinylation of immunopurified endogenous POLRMT (I), mtDNA-encoded gene expression (J), MTG intensity (K), and oxygen consumption (L) were determined. Data represent means ± SD, $n = 3$ independent biological replicates, $t$ test. **$P < 0.01$; *$P < 0.05$; n.s. not significant. (M) Correlation analysis of *MRPL45* and *SUCLG1* mRNA levels in cancer cell line encyclopedia. (N) Relative ATP and GTP abundance in indicated cell lines were quantified and normalized to total protein. Data represent means ± SD, $n = 3$ independent biological replicates. (O) K622 succinylation of immunopurified endogenous POLRMT was determined in indicated cell lines. Source data are available online for this figure.

Depletion of *SUCLG1* resulted in hypersuccinylation of immunopurified POLRMT, which was accompanied by deficient POLRMT binding with TFAM and TFB2M (Figs. 3B and EV3F). Notably, the succinylation of POLRMT mutants was minimally altered after depleting *SUCLG1* (Figs. 3B and EV3F). These results suggest that SUCLG1 restricts K622 succinylation of POLRMT to maintain the association of the transcription complex.

The sugar-phosphate backbone of DNA carries negative charges, which provides a structural basis for the positive-charged K622 to mediate POLRMT binding of mtDNA. We speculated that K622 succinylation would disrupt POLRMT-mtDNA association and suppress mitochondrial transcription. To test it, we carried out mitochondrial genome immunoprecipitation (MitoChIP) assay to determine the interaction between POLRMT and mtDNA. Wild-type POLRMT and its K622R mutant remarkably interacted with mtDNA (Figs. 3C and EV3G). POLRMT^K622E, which mimics the succinylated state, showed impaired binding with mtDNA (Figs. 3C and EV3G). Importantly, mtDNA binding of wild-type POLRMT, but not its K622 mutants, was reduced in *SUCLG1*-knockdown cells (Figs. 3C and EV3G). Next, we performed in organello transcription assay with radiolabeled UTP to directly monitor POLRMT activity. Compared to wild-type POLRMT, K622E mutant is insufficient to maintain UTP radiolabeling in mitochondria isolated from stable MV411 cells (Fig. 3D,E). POLRMT^K622R mutant rescued mitochondrial transcription to a level similar to wild-type POLRMT, supporting the regulatory role of K622 succinylation in mitochondrial transcription (Fig. 3D,E). Depletion of *SUCLG1* strongly reduced mitochondrial transcription in cells rescued by POLRMT^WT, but to a lesser extent in those rescued by K622E mutant (Fig. 3D,E). In agreement, silencing *SUCLG1* efficiently decreased mtDNA abundance, mtDNA-encoded gene expression, and oxygen consumption in cells re-expressing POLRMT^WT, but to a lesser extent in cells rescued by K622 mutants (Fig. 3F–H). These results collectively demonstrate that SUCLG1 represses K622 succinylation of POLRMT to maintain mitochondrial transcription.

The signaling role of succinyl-CoA indicates the involvement of other TCA cycle enzymes in mitochondrial transcription. Genetic mutations of succinate dehydrogenase (SDH), the enzyme downstream of SUCLG1, were found in a variety of human malignancies (Sulkowski et al, 2018). We stably depleted *SDHA* and observed a significant accumulation of succinate and succinyl-CoA in MV411 cells, accompanied by increased K622 succinylation of endogenous POLRMT (Fig. EV3H,I). In line with these findings, both mtDNA copy number and MTG intensity were decreased in

*SDHA*-knockdown cells (Fig. EV3J,K). It is thus possible that SDH contributes to tumor development by regulating mitobiogenesis.

SUCLG1-mediated succinyl-CoA metabolism plays dual roles, through restricting POLRMT succinylation and sustaining TCA cycle, to contribute to mitochondrial function. To define the relationship between different roles of SUCLG1, we knocked down *OGDH* to reduce succinyl-CoA level in MV411 cells (Figs. 3I and EV3L). Next, cell-permeable succinate (dimethyl succinate, DMS) was replenished to stable cells to ensure the proper functioning of TCA cycle (Fig. EV3M). This cell-based model allows us to specifically define how succinyl-CoA signaling regulates mitochondrial transcription. Loss of *OGDH* reduced K622 succinylation of POLRMT, which was minimally altered by DMS treatment (Fig. 3I). However, depleting *OGDH* mildly altered mtDNA-encoded gene expression and mitochondrial mass (Fig. 3J,K). Interestingly, mitochondrial respiration was decreased upon OGDH knockdown (Fig. 3L). Supplementation with DMS did not show a significant impact on mtDNA-encoded gene expression, mitochondrial mass, and respiration in control cells, but remarkably enhanced mitobiogenesis and respiration in *OGDH*-knockdown cells (Fig. 3J–L). These results indicate that POLRMT hyposuccinylation failed to elevate mitochondrial activity with a defective TCA cycle. POLRMT hyposuccinylation is necessary, but not sufficient, for mitobiogenesis. Importantly, some of the phenotypes observed with *SUCLG1* depletion may be related to the loss of its canonical function. Both the enzymatic activity of SUCLG1 in the TCA cycle and SUCLG1-mediated hyposuccinylation of POLRMT are essential for mitochondrial function.

SUCLG1 catalyzes the substrate-level phosphorylation of either ADP or GDP, which implies that cells with high rates of ATP or GTP synthesis may have POLRMT hyposuccinylation and increased mitobiogenesis. Mitochondrial protein translation is a highly energy-consuming process that requires both ATP and GTP (Kummer and Ban, 2021). We investigated the potential link between POLRMT succinylation and mitochondrial translation. MRPL45 is a key factor in mitochondrial ribosomes that determines protein translational activity (Greber et al, 2014). Transcriptomes of cancer cell line encyclopedia (Ghandi et al, 2019) revealed that *SUCLG1* expression showed a significantly positive correlation with *MRPL45* (Fig. 3M). We adopted *MRPL45*^low (Hs578T and HCC1806) and *MRPL45*^high (KM12 and A549) cell lines for further analysis. Nucleotide quantification revealed that *MRPL45*^high cell lines had higher levels of ATP and GTP, which were accompanied by decreased POLRMT K622 succinylation (Fig. 3N,O). The link between POLRMT

succinylation and energy production potentially couples mitochondrial transcription and translation to ensure efficient production of mtDNA-encoded proteins.

## Leukemia-derived FLT3 mutants upregulate SUCLG1 to boost mitobiogenesis

AML is a genetically heterogeneous malignancy with enhanced mitobiogenesis, resulting in its dependence on mitochondrial respiration (Li et al, 2016). To identify the genetic factor(s) that remodel mitobiogenesis in AML, we tested POLRMT succinylation in a panel of AML cell lines (Fig. 4A). Human CD34+ CB cells from three different healthy donors were included as normal controls. Endogenous POLRMT was immunoprecipitated from various cell lines and K622 succinylation was determined by western blotting. Interestingly, SUCLG1 protein expression was negatively linked to K622 succinylation in AML cell lines (Fig. 4B). We further extracted metabolites from tested cell lines and quantified cellular succinyl-CoA, which tended to be positively correlated with POLRMT succinylation (Fig. 4C). FLT3 is a membrane-bound tyrosine kinase that is mutated in ~30% of AML patients (Antar et al, 2020). In *FLT3*-mutated cell lines (MONOMAC6, MOLM14, and MV411), SUCLG1, but not SDHA, succinate dehydrogenase subunit B (SDHB), fumarate hydratase (FH), SIRT5 or POLG, was overexpressed (Fig. 4A). In accordance, K622 succinylation levels were lower in *FLT3*-mutated cells compared to that in other AML cell lines and healthy controls (Fig. 4A). These results point to the possibility that FLT3 remodels mitobiogenesis in AML cells.

Mutations of FLT3 at the tyrosine kinase domain (TKD) or internal tandem duplication (ITD) mutations result in its constitutive activation (Daver et al, 2021). To test whether FLT3 modulates mitobiogenesis, we overexpressed HA-tagged FLT3 and its mutants into CD34+ CB cells. FLT3$^{TKD}$ and FLT3$^{ITD}$ mutants, but not the wild-type kinase, enhanced SUCLG1 expression and decreased succinyl-CoA level (Fig. 4D,E). In agreement, POLRMT succinylation was reduced by overexpressing FLT3 mutants (Fig. 4D). mtDNA abundance, mtDNA-encoded gene expression, mitochondrial mass and cellular respiration were simultaneously increased (Figs. 4F–H and EV4A). Meanwhile, enzymatic activities of SDH, but not FH, were elevated by mutant FLT3 (Fig. EV4B,C). Treatment of CD34+ CB cells with a chemical inhibitor against SIRT5 upregulated POLRMT succinylation and reduced mtDNA binding of POLRMT, which is accompanied by decreased expression of mtDNA-encoded genes (*MT-CO1*, *MT-ND6*, and *MT-ATP6*) (Fig. EV4D–F). In FLT3$^{ITD}$-expressing cells, SIRT5 inhibition altered POLRMT succinylation and mtDNA binding to a lesser extent (Fig. EV4D,E). A similar pattern was observed in the mRNA expression of mtDNA-encoded genes (Fig. EV4F). Lysine succinylation of endogenous POLG remained unaltered in FLT3$^{ITD}$-expressing CD34+ CB cells (Fig. EV4G). We further depleted *SUCLG1* in stable CD34+ CB cells and tested mitochondrial respiration. FLT3$^{ITD}$ markedly increased oxygen consumption and respiration complexes activities in a SUCLG1-dependent manner (Fig. 4I–K).

Next, we tested the regulation of SUCLG1 by FLT3 in human leukemia cells. Genetic or chemical inhibition of *FLT3* reduced SUCLG1 expression, increased succinyl-CoA level and K622 succinylation of immunopurified POLRMT in MV411 and MOLM14 cells (Figs. 4L,M and EV4H–M). More importantly,

mtDNA abundance, mtDNA-encoded gene expression, and mitochondrial mass were decreased in *FLT3*-knockdown MV411 cells (Fig. 4N–P). Further, we evaluated whether *FLT3* mutation status was linked to mRNA expression of mitochondrial genes in TCGA AML dataset. *FH* and *POLG*, but not the other tested genes, showed enhanced mRNA expression in *FLT3*-mutated samples (Fig. EV4N). Interestingly, treatment of MV411 cells with FLT3 inhibitors (FLT3-IN-3 and AC220) had a negligible effect on protein expression of SDH, FH, SIRT5, and POLG (Fig. EV4O). Enzymatic activities of SDH, but not FH, were decreased in FLT3 inhibitor-treated MV411 cells (Fig. EV4P,Q). These observations suggest that SDH expression and activity may be differentially regulated by SUCLG1 and FLT3 signaling. Lysine succinylation of endogenous POLG also showed no obvious changes after chemical inhibition of FLT3 (Fig. EV4R). In addition, we co-expressed shRNAs against *FLT3* and *SUCLG1* into MV411 cells (Fig. EV4S). While silencing either gene decreased oxygen consumption and respiration complex activities, co-depletion of both genes did not result in a further decrease in mitochondrial respiration (Fig. 4Q,R). FLT3 signaling may have specificities toward different respiration complexes as complex IV activity was not significantly altered in *FLT3*-knockdown cells (Fig. 4R). Nevertheless, these data collectively demonstrate that mutant FLT3 upregulates SUCLG1 to mediate POLRMT hyposuccinylation and boost mitobiogenesis.

## FLT3 signaling modulates nuclear transcription to enhance mitochondrial respiration

Mutant FLT3 upregulates mitochondrial transcription and mitobiogenesis, which implies that FLT3 signaling controls a more general set of mitochondrial genes in the nucleus. The impact of FLT3 on leukemia transcriptome has been previously profiled in MOLM14 and MV411 cells (Sabatier et al, 2023; Data ref: Sabatier et al, 2023b). We collected the differentially expressed mitochondrial genes (Fig. 5A,B), among which 25 genes were strongly downregulated (Fold change ≥2; *P* < 0.01) in both cell lines after depleting *FLT3* (Fig. 5C). We validated that mRNA expression of these genes was decreased in *FLT3*-knockdown MV411 cells (Fig. 5D). mRNA expression of *citrate synthase* (*CS*), which serves as a negative control, was not significantly altered in *FLT3*-knockdown cells (Fig. 5D). Analysis of potential promoter regions of these genes revealed at least three different consensus sequences (Figs. 5E–G and EV5A–C), which potentially allows transcription regulators to jointly regulate their expression in response to FLT3 signaling.

Focusing on SUCLG1, we set out to define its transcriptional regulation by FLT3. FLT3 was previously reported to control at least 16 different transcription factors or regulators. We established a shRNA-library against the known transcription regulators (three shRNAs each) and transduced them into MV411 cells. Depleting *E2F1*, but not the other transcription factors, strongly suppressed *SUCLG1* expression (Fig. 5H). *SUCLG1* expression level was also dramatically decreased in *E2F1*-knockdown MOLM14 cells (Fig. EV5D,E), further supporting E2F1 as a potential governor of *SUCLG1* transcription. SUCLG1 protein was downregulated in *E2F1*-depleted MV411 cells, which was coupled with hypersuccinylation of POLRMT (Fig. 5I). Treatment of *E2F1*-knockdown cell with FLT3 inhibitor did not further elevate POLRMT succinylation (Fig. 5I). Next, we examined E2F1 binding with the *SUCLG1* gene.

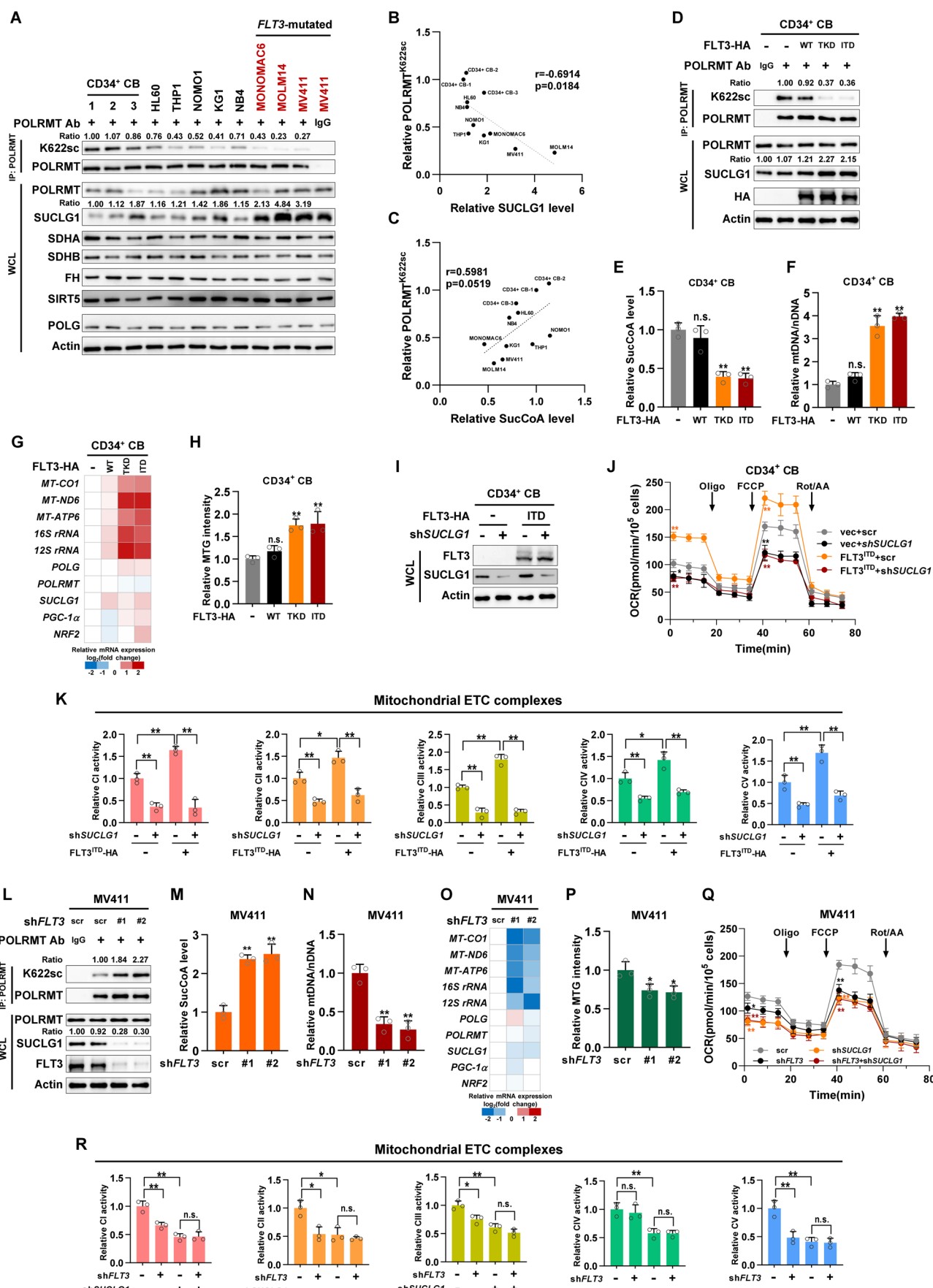

**Figure 4.  Leukemia-derived FLT3 mutants upregulate SUCLG1 to boost mitobiogenesis.**

(A) A panel of AML cells was subjected to western blotting to determine protein expression. CD34[+] CB cells from three different donors were included as normal controls. K622 succinylation (K622sc) of immunopurified endogenous POLRMT was assayed. AML cell lines carrying *FLT3* mutations were colored red. IgG was included as a negative control for immunoprecipitation. Ratios of K622sc and SUCLG1 were normalized to POLRMT protein and Actin, respectively. (B, C) Metabolites were extracted to determine cellular succinyl-CoA levels. The correlation of POLRMT[K622sc] with SUCLG1 protein (B) and succinyl-CoA abundance (C) were analyzed. Pearson's correlation. (D–H) HA-tagged FLT3 and its mutants were stably expressed in CD34[+] CB cells. Endogenous POLRMT was immunopurified to determine K622 succinylation. Ratios of K622sc were normalized to POLRMT protein (D). Cellular succinyl-CoA levels (E), mtDNA abundance (F), mtDNA-encoded gene expression (G), and MTG intensity (H) were determined. Data represent means ± SD, $n = 3$ independent biological replicates, *t* test. **$P < 0.01$; n.s. not significant. (I–K) shRNA against *SUCLG1* (#1) was stably expressed in control or FLT3[ITD]-expressing cells (I). Oxygen consumption (J) and activities of ETC complexes (K) were determined. Data represent means ± SD, $n = 3$ independent biological replicates, *t* test. **$P < 0.01$; *$P < 0.05$. (L–P) shRNAs against *FLT3* were stably expressed in MV411 cells. Endogenous POLRMT was immunopurified to determine K622 succinylation. Ratios of K622sc were normalized to POLRMT protein (L). Cellular succinyl-CoA levels (M), mtDNA abundance (N), mtDNA-encoded gene expression (O), and MTG intensity (P) were determined. Data represent means ± SD, $n = 3$ independent biological replicates, *t* test. **$P < 0.01$; *$P < 0.05$. (Q, R) Scrambled control or shRNAs targeting *SUCLG1* (#1) or *FLT3* (#1) were co-expressed in MV411 cells. Oxygen consumption rates (Q) and activities of ETC complexes (R) in stable cells were determined. Data represent means ± SD, $n = 3$ independent biological replicates, *t* test. **$P < 0.01$; *$P < 0.05$; n.s. not significant. Source data are available online for this figure.

The genomic DNA sequence around the transcription start site (TSS) of *SUCLG1* contains at least two different E2F1-binding motifs (Fig. 5J). We performed ChIP-qPCR assay with E2F1-specific antibody. The occupancy of E2F1 at *SUCLG1* gene was readily detected in both MV411 and MOLM14 cells (Fig. 5K,L). E2F1 binding of *SUCLG1* was strongly weakened in *E2F1*-knockdown cells, supporting the specificity of the ChIP antibody (Fig. 5K,L). More importantly, *E2F1* depletion suppressed cellular respiration of MV411. Treatment with FLT3 inhibitor did not further reduce oxygen consumption in *E2F1*-knockdown cells (Fig. 5M). Collectively, FLT3 signaling modulates nuclear transcription programs to enhance mitochondrial respiration.

## SUCLG1 suppresses POLRMT succinylation to support leukemic proliferation

Given that mitochondrial respiration is indispensable to leukemic growth (Egan et al, 2021), we investigated the pathophysiological function of POLRMT succinylation in leukemia cells. We knocked down *SUCLG1* in MV411 cells and observed a marked suppression of cell proliferation (Fig. 6A). Re-expression of SUCLG1[WT], but not its catalytic mutants, restored cell growth (Figs. 6A,B and EV6A). Treatment with POLRMT-specific inhibitor strongly reduced cell proliferation and colony formation in cells rescued by SUCLG1[WT], but failed to further decrease the proliferation of cells re-expressing SUCLG1 mutants (Figs. 6B and EV6A). Similar results were observed after treating *SUCLG1*-knockdown and rescue cells with FLT3 inhibitor (Fig. EV6B,C). These data suggest that FLT3 and SUCLG1 work in collaboration with POLRMT to regulate leukemic proliferation. We further assessed the proliferation of *POLRMT*-knockdown and rescue cells. Re-expression of POLRMT[K622R], but not POLRMT[K622E], restored cell proliferation (Fig. 6C). To examine the contribution of POLRMT succinylation to cell proliferation, we depleted *SUCLG1* in *POLRMT*-knockdown and rescue cells. Loss of *SUCLG1* suppressed the proliferation of POLRMT[WT]-rescued cells, but not the POLRMT[K622E]-rescued counterparts (Figs. 6D and EV6D). Notably, loss of *SUCLG1* also demonstrated a growth-suppressive effect in cells re-expressing POLRMT[K622R] (Figs. 6D and EV6D), suggesting that the relationship between SUCLG1 and POLRMT in the context of leukemia is not linear. SUCLG1 may have functions other than modulating POLRMT succinylation to support leukemia cell proliferation.

We hypothesized that enhanced mitobiogenesis may confer a dependency on oxidative phosphorylation to *FLT3*-mutated cells. As anticipated, cell lines carrying *FLT3* mutations displayed increased sensitivities to rotenone, an ETC complex I inhibitor, in cell survival assay (Fig. 6E). Besides, overexpressing FLT3[ITD] elevated the sensitivity of CD34[+] CB cells to rotenone (Fig. 6F).

We next tested leukemia development in vivo. FLT3[ITD] alone is insufficient to drive leukemia formation, we introduced FLT3[ITD] to MLL-AF9-driven leukemia model as previously reported (Stubbs et al, 2008). HA-tagged FLT3[ITD] was co-transduced with MLL-AF9 into mouse bone marrow cells to establish mouse leukemia (Fig. 6G). Further, shRNA against *Suclg1* or *Polrmt* was stably expressed in leukemic bone marrow cells. Stable bone marrow cells were transplanted to sublethally irradiated recipients (Figs. 6H and EV6E). Depleting either *Suclg1* or *Polrmt* significantly prolonged animal survival (Figs. 6I and EV6F). In line with the results obtained from cultured cell lines, *Suclg1*-knockdown group had intermediate survival in POLRMT[K622R] mutant setting. This observation also suggests additional functions of SUCLG1 enzyme activity or succinylation may exist to impact leukemia development. Together, SUCLG1 and POLRMT are essential for mutant *FLT3*-driven leukemia.

Next, we established a humanized leukemia model by transplanting stable MV411 cells into sublethally irradiated NSG mice. Re-introduction of wild-type POLRMT or POLRMT[K622R] in *POLRMT*-knockdown cells restored leukemia progression to a similar level (Fig. 6J). Depleting *SUCLG1* in MV411 cells re-expressing POLRMT[WT] significantly delayed leukemia development, while a milder inhibitory effect was observed in leukemia model established with POLRMT[K622R] mutant (Fig. 6J). Besides, chemical inhibition of FLT3 prolonged animal survival to a lesser extent in POLRMT[K622R] group compared to POLRMT[WT] group (Fig. EV6G). These data show that POLRMT succinylation is a major target, if not the sole, for mutant FLT3 to promote leukemia progression.

To determine the pathological relevance of POLRMT succinylation, human normal and leukemic bone marrow samples were collected and sequenced to determine *FLT3* mutation status. In total, 15 different leukemic samples were included in our study ($n = 5$ for *FLT3*[WT] AML; $n = 10$ for AML with *FLT3*[TKD/ITD]). Protein expression levels in whole-cell lysate and K622 succinylation of immunopurified POLRMT were determined (Fig. EV6H).

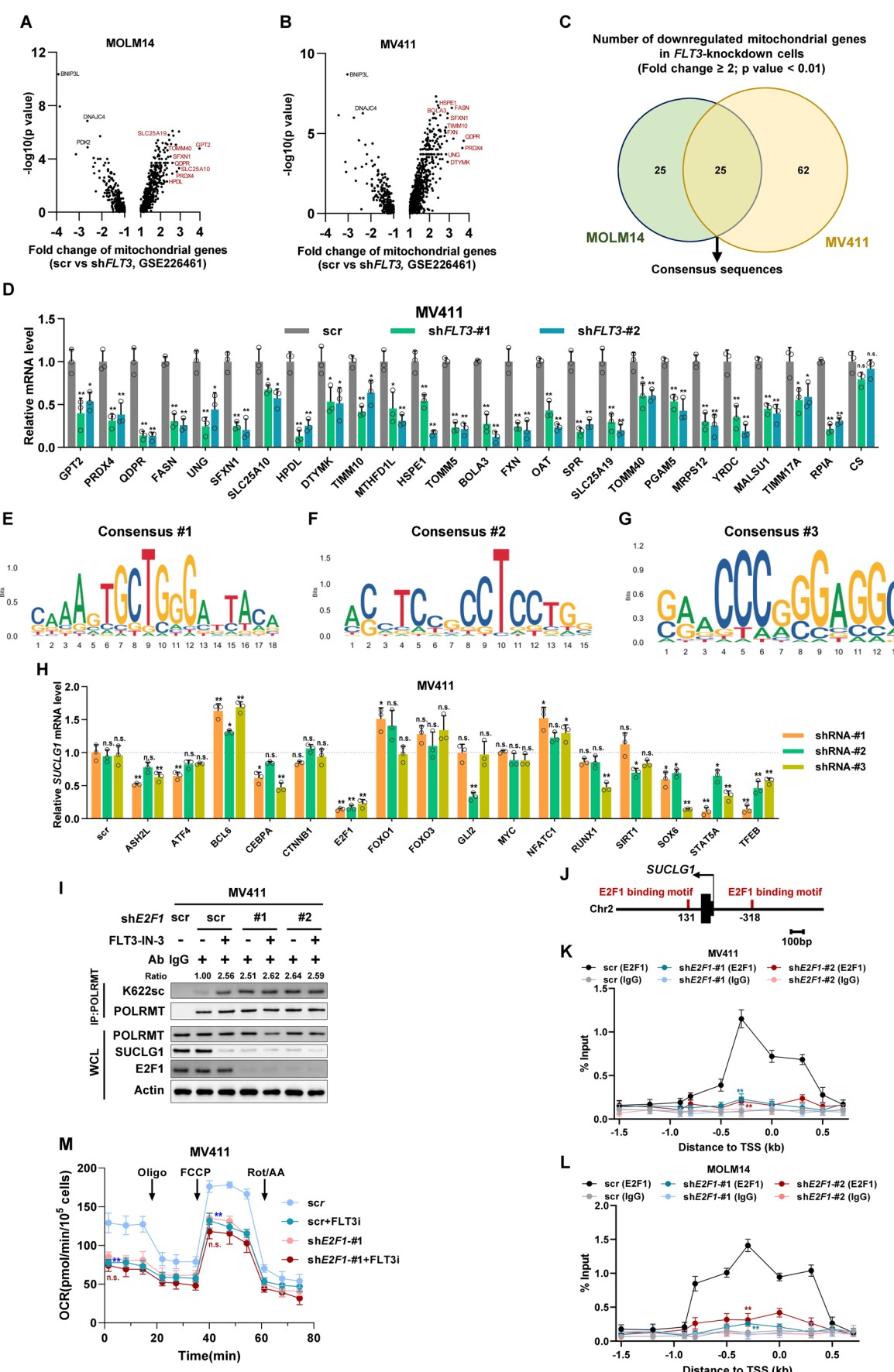

**Figure 5.  FLT3 signaling modulates nuclear transcription to enhance mitochondrial respiration.**

(A, B) Shown are differentially expressed mitochondrial genes in control and *FLT3*-knockdown cells. Data were extracted from GSE226461, *n* = 3 independent biological replicates, *t* test. (C) Venn diagram showing the numbers of significantly downregulated mitochondrial genes in MV411 and MOLM14 cells after depleting *FLT3*, *n* = 3 independent biological replicates, *t* test. (D) mRNA expression of 25 overlapped genes in (C) was validated in *FLT3*-knockdown MV411 cells. *CS* was included as a negative control. Data represent means ± SD, *n* = 3 independent biological replicates, *t* test. **P* < 0.01; **P* < 0.05; n.s. not significant. (E–G) Shown are the consensus sequences in the promoter regions of 25 mitochondrial genes regulated by FLT3. (H) shRNAs targeting *FLT3*-regulated transcription factors were transduced into MV411 cells. *SUCLG1* mRNA levels were determined by qPCR. Data represent means ± SD, *n* = 3 independent biological replicates, *t* test. **P* < 0.01; **P* < 0.05; n.s. not significant. (I) Scrambled control or *E2F1*-knockdown MV411 cells were treated with or without FLT3 inhibitor (FLT3-IN-3). K622 succinylation of endogenous POLRMT was determined by western blotting. (J) Schematic overview of *SUCLG1* promoter and TSS region. (K, L) Control IgG or E2F1 antibodies were used for chromatin immunoprecipitation. Scrambled control or *E2F1*-knockdown MV411 or MOLM14 cells were subjected to ChIP-qPCR assay. Data represent means ± SD, *n* = 3 independent biological replicates, *t* test. **P* < 0.01. (M) Scrambled control or *E2F1*-knockdown MV411 cells were treated with or without FLT3 inhibitor (FLT3-IN-3). Oxygen consumption rates were determined. Data represent means ± SD, *n* = 3 independent biological replicates, *t* test was performed between untreated and FLT3i-treated groups. **P* < 0.01; n.s. not significant. Source data are available online for this figure.

Metabolite quantification revealed that succinyl-CoA levels were reduced in both $FLT3^{WT}$ and $FLT3^{TKD/ITD}$ AML samples, compared to the healthy controls (Fig. 6K). K622 succinylation was remarkably decreased in *FLT3*-mutated AML, while no significant changes in POLRMT protein expression were observed (Figs. 6L and EV6I). In line with this result, SUCLG1 protein was overexpressed in *FLT3*-mutated AML samples (Fig. 6M). Importantly, *FLT3*-mutated AML samples had increased mtDNA copies and higher *MT-CO1* mRNA expression which indicated enhanced mitobiogenesis (Fig. EV6J,K). We further investigated the relationship between SUCLG1 and leukemia development in TCGA AML dataset. mRNA levels of *SUCLG1*, but not *POLRMT*, were remarkably elevated in patients carrying *FLT3* mutants (Fig. 6N,O). Strikingly, higher expression of *SUCLG1* was linked to worse overall survival in $FLT3^{WT}$ patients, but not in $FLT3^{TKD/ITD}$ individuals (Fig. 6P). This observation also implies that additional functions of FLT3 and SUCLG1 may exist to regulate mitochondrial biogenesis and contribute to leukemia development. In summary, SUCLG1 represses succinyl-CoA levels to restrict POLRMT succinylation and enhance mitochondrial activity. Mutant *FLT3* modulates nuclear transcription programs to upregulate *SUCLG1* expression and mitochondrial genes, resulting in mitobiogenesis remodeling and AML progression (Fig. 6Q).

## Discussion

In the nucleus, metabolites were found to act as signaling molecules that regulate epigenetics and gene transcription (Sullivan et al, 2016). Whether metabolites regulate mitochondria-specific biogenesis programs remains to be defined. In this study, we found that TCA cycle intermediate succinyl-CoA functions as a signaling molecule to modify and negatively regulate POLRMT. The transcription of mtDNA and nDNA-encoded ETC genes is highly coordinated, it is reasonable to speculate that unknown succinyl-CoA signaling pathway might exist in the nucleus to modulate mitochondrial gene expression.

Succinyl-CoA serves as a succinyl group donor to modify numerous proteins (Frezza, 2017), the majority of which are located in mitochondria (Yang and Gibson, 2019). Here we found that SUCLG1 constrained succinyl-CoA levels to inhibit the POLRMT succinylation and impede POLRMT binding to mtDNA. *SUCLG1* showed positive ETC correlations in most types of cancers. To be noted, genetic mutations of different subunits of human succinate-CoA ligase have been implicated in mtDNA depletion syndrome

(Elpeleg et al, 2005; Hadrava Vanova et al, 2022; Ostergaard et al, 2007). Although *SUCLA2* and *SUCLG2* didn't show a consistent correlation with ETC gene expression in TCGA transcriptomes, they may be regulated at post-transcriptional levels to connect succinyl-CoA with mtDNA function. SUCLG2 and SUCLA2 primarily carry out anabolic and catabolic functions, respectively (Gut et al, 2020). Inherent genetic mutations of succinyl-CoA ligase predominantly targeted *SUCLA2* and *SUCLG1* (Hadrava Vanova et al, 2022). Disease-causing mutations of *SUCLG2* are rarely reported and their role in mtDNA transcription remains obscure (Carrozzo et al, 2016). This intriguing phenomenon may be explained in different ways: (1) POLRMT succinylation is intimately linked to succinate-CoA-mediated ATP production, which is majorly present in highly energy-consuming organs where catabolism primarily occurs; (2) it is possible that SUCLA2 could compensate for the loss of SUCLG2, but not vice versa. Further efforts are required to elucidate the functional diversifications of succinate-CoA ligase subunits in mitochondria metabolism.

Protein succinylation is a dynamic modification. The functional interaction between SIRT5 and POLRMT is currently under investigation. It is worthy to note that various types of PTM have been identified in POLRMT, such as phosphorylation and arginine methylation (Hornbeck et al, 2015). It would not be surprising to expect the involvement of more metabolite signals in regulating POLRMT activity. Interaction of metabolic enzymes with their binding partners may ensure specific and efficient regulation of downstream events. We found that SUCLG1 readily associated with POLRMT but not POLG. Although this observation does not exclude the possibility that SUCLG1 modulates POLG modification, SUCLG1 may more efficiently modulate the succinylation of substrate proteins through direct physical interaction.

Cancer-promoting mutations not only reshape cell metabolism but also remodel mitobiogenesis. The mechanism of how tumor cells remodel mitobiogenesis is poorly understood. Previous studies have shown that $FLT3^{ITD}$ upregulated glutamine metabolism to promote leukemia progression (Gallipoli et al, 2018). Besides, $FLT3^{ITD}$ upregulated mitochondrial metabolism and oxidative phosphorylation in AML (Erdem et al, 2022a, 2022b). We here demonstrated that mutant *FLT3* suppressed succinyl-CoA signaling to enhance mitobiogenesis and promote cancer development. Whether additional oncogenic mutations reprogram mitobiogenesis requires further exploration. Importantly, SUCLG1 is a key regulator of mtDNA replication and transcription, targeting SUCLG1 would be beneficial in treating leukemia and solid tumors that are highly dependent on mitochondria. Besides, it would be

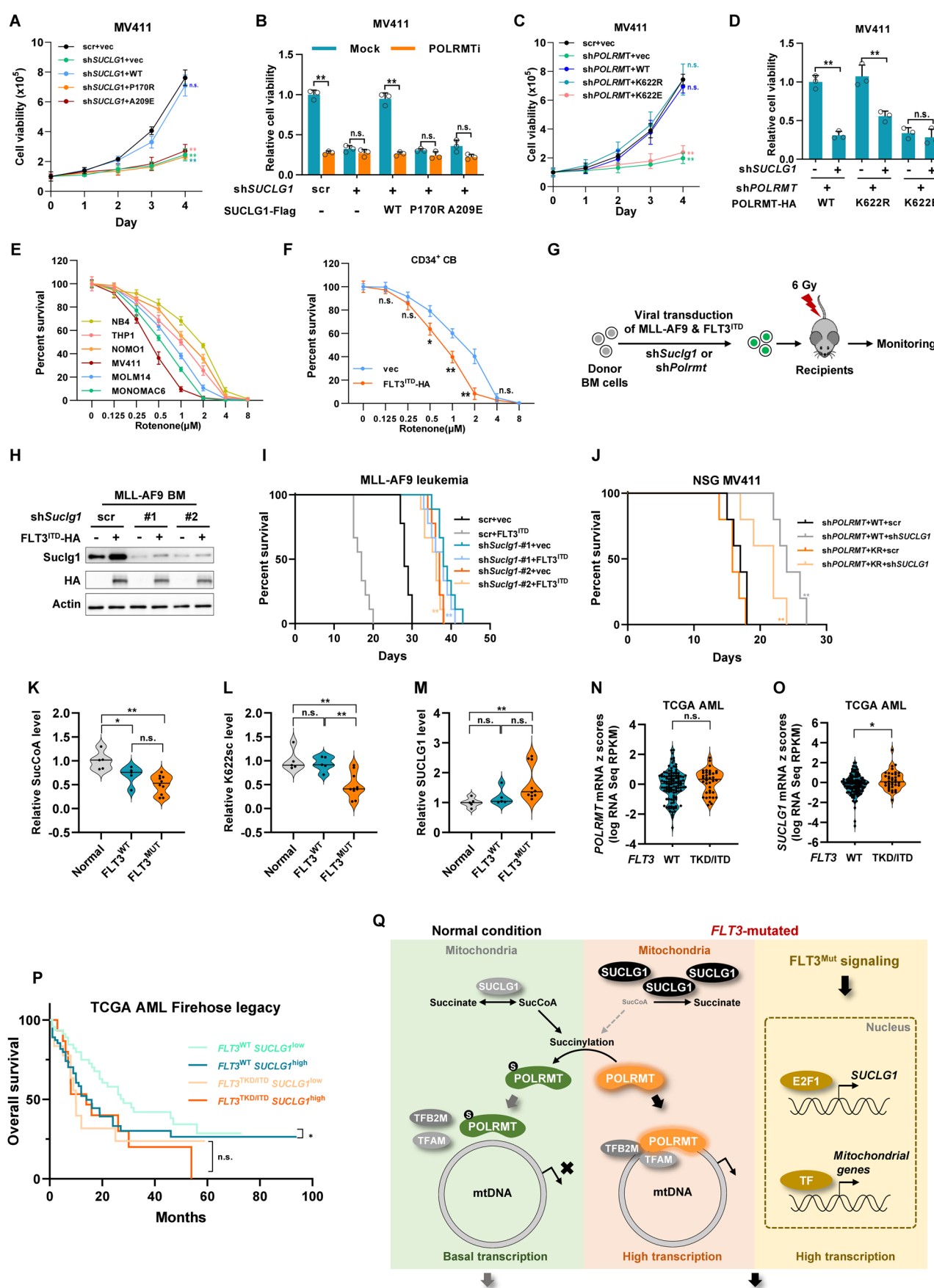

◄   **Figure 6.   SUCLG1 suppresses POLRMT succinylation to support leukemic proliferation.**

(A, B) *SUCLG1*-knockdown and rescue MV411 cells were subjected to cell proliferation assay (A). Cells were further treated with POLRMT inhibitor (POLRMTi) to determine viability (B). Data represent means ± SD, $n = 3$ independent biological replicates, $t$ test. **$P < 0.01$; n.s. not significant. (C, D) *POLRMT*-knockdown and rescue MV411 cells were subjected to cell proliferation assay (C). shRNA targeting *SUCLG1* (#1) was stably expressed to determine cell proliferation (D). Data represent means ± SD, $n = 3$ independent biological replicates, $t$ test. **$P < 0.01$; n.s. not significant. (E, F) Leukemia cells or CD34+ CB cells were cultured with increasing doses of rotenone as indicated. Cell viability was determined after three days of culture. Data represent means ± SD, $n = 3$ independent biological replicates, $t$ test. **$P < 0.01$; *$P < 0.05$; n.s. not significant. (G) Schematic overview showing the establishment of mouse MLL-AF9 + FLT3^ITD leukemia. (H, I) MLL-AF9 bone marrow cells were transduced with vector control or HA-tagged FLT3^ITD. Scrambled control or shRNAs targeting *Suclg1* was stably expressed. Protein expression was determined by western blotting (H). Animal survival was assayed after transplanting leukemic bone marrow (I). $n = 9$ mice, log-rank (Mantel–Cox) test. **$P < 0.01$. (J) *POLRMT*-knockdown and rescue MV411 cells were transduced with scrambled control or shRNA against *SUCLG1*. Cells were transplanted into sublethally irradiated NSG mice. Animal survival was determined. $n = 5$ mice, log-rank (Mantel–Cox) test. **$P < 0.01$. (K–M) Human clinical AML bone marrow samples were collected. Metabolites were extracted to determine succinyl-CoA abundance (K). K622 succinylation of immunopurified POLRMT (L) and SUCLG1 protein expression (M) were determined by western blotting. $n = 5$–10, $t$ test. **$P < 0.01$; *$P < 0.05$; n.s. not significant. (N–P) mRNA expression of *POLRMT* (N) and *SUCLG1* (O) in *FLT3*^WT ($n = 116$) and *FLT3*^TKD/ITD ($n = 38$) TCGA AML samples were analyzed. $t$ test. *$P < 0.05$; n.s. not significant. TCGA AML patients were grouped by *FLT3* mutation status and *SUCLG1* mRNA expression level. Overall survival was determined (P). log-rank (Mantel–Cox) test. *$P < 0.05$; n.s. not significant. (Q) Working model of SUCLG1-mediated succinyl-CoA signaling and FLT3-induced remodeling of mitobiogenesis. Source data are available online for this figure.

interesting to understand whether other metabolic enzymes modulate mitochondria-specific biogenesis programs to contribute to tumorigenesis and cancer development.

# Methods

## Cell culture and treatment

All cells were cultured at 37 °C under 5% $CO_2$ humidified atmosphere. Human embryonic kidney cell line (HEK293T), human AML cell lines (HL60, KG1, MOLM14, MONOMAC6, MV411, NB4, NOMO1, THP1), colorectal cancer cell lines (HCT116, KM12), lung cancer cell line (A549), breast cancer cell lines (Hs578T, HCC1806), and liver cancer cell line (HepG2) were maintained in RPMI 1640 medium or Dulbecco's modified Eagle medium (DMEM) (Invitrogen) supplemented with 10% fetal bovine serum (Biological Industries) in the presence of penicillin, streptomycin, and 2 mM glutamine (Corning). Human cord blood (CB) CD34+ cells were grown in RPMI 1640 (with serum and glutamine) supplemented with recombinant human growth factors, including 40 ng/mL IL-6 (Peprotech), 50 ng/mL FLT3 ligand (Peprotech), 20 ng/mL stem cell factor (SCF) (Peprotech), and 50 ng/ml TPO (Peprotech). Mouse BM cells were maintained in RPMI 1640 (with serum and glutamine) supplemented with recombinant murine growth factors, including 10 ng/mL IL3 (R&D systems), 10 ng/mL SCF (R&D Systems), 100 ng/mL IL-6 (Peprotech). Cells were treated with SIRT5 inhibitor (SIRT5i, 0.5 μM, MedChemExpress #HY-112634), FLT3-IN-3 (1 nM, MedChemExpress #HY-112145), AC220 (1 nM, MedChemExpress #HY-13001), dimethyl succinate (50 μM, Sigma), uridine (0.2 mM, Sigma) and pyruvate (1 mM, Sigma) for the indicated duration.

## Plasmids and antibodies

The cDNAs encoding full-length human SUCLG1, POLRMT, FLT3, and Ubiquitin were cloned into Flag, HA, Myc or His-tagged vectors (pcDNA3.1, pSJ3, and pLV-EF1a-IRES). All expression constructs were verified by DNA sequencing. Plasmid transfection was carried out by using polyethylenimine or Lipofectamine 3000 (ThermoFisher). Antibodies against MT-CO1 (Novus biologicals

#NBP2-29949), MT-ND6 (Novus biologicals #NBP1-70650), MT-ATP6 (Sigma #MABS1995), SDHA (Cell Signaling #11998), NDUFB8 (Novus biologicals #NBP2-75586), POLG (Abcam #ab128899), POLRMT (Abcam #ab167368), MRPL45 (Sigma #HPA023373), PGC-1α (Abcam #ab54481), NRF2 (Abcam #ab89443), beta-actin (Cell Signaling #3700), Flag (Sigma #F1804), SUCLG1 (Abcam #ab97867), TFAM (Cell Signaling #8076), suc-Lys (Novus biologicals #NBP3-16032), HA (Santa Cruz #sc-7392), HSP60 (Cell Signaling #4870), G6PD (Abcam #ab133525), Histone H3 (Cell Signaling #3638), me-Lys (Cell Signaling #14679), ac-Lys (Cell Signaling #9441), p-Tyr (Santa Cruz #sc-508), SUCLG2 (Santa Cruz #sc-390818), SUCLA2 (Novus biologicals #NBP1-33015), His (Cell Signaling #2366), TFB2M (Cell Signaling #50837), FLT3 (Cell Signaling #3462), Myc tag (Cell Signaling # 2276), OGDH (Abcam #ab137773), Ubiquitin (Abcam #ab134953), SDHB (Abcam #ab175225), FH (Abcam #ab233394), SIRT5 (Abcam #ab259967), POLG (Abcam #ab128899), E2F1 (Santa Cruz #sc-251) were purchased commercially. To generate a site-specific antibody to detect the succinyl-lysine 622 of POLRMT [α-K622sc], synthesized peptide QIGILK(sc)PHPAY (GL Biochem) was coupled to KLH as an antigen to immunize rabbit. Anti-serum was collected after four doses of immunization.

## Immunoprecipitation and western blotting

Cells were lysed in ice-cold NP-40 buffer [50 mM Tris-HCl (pH 7.4), 150 mM NaCl, 0.3% NP-40] containing protease inhibitor cocktail (Sigma). Immunoprecipitation was carried out either by incubating Flag beads (Sigma) at 4 °C with lysate for three hours or by incubating HA, Myc tag, SUCLG1, POLG and POLRMT antibodies with cell lysate for one hour, followed by incubating with Protein-A beads (Millipore) for another two hours at 4 °C. After the incubation, beads were washed three times with ice-cold NP-40 buffer. Standard western blotting protocols were adopted.

## Mitochondrial chromatin immunoprecipitation and quantitative PCR (MitoChIP-qPCR)

Briefly, cells were resuspended in mitochondria isolation buffer (MIB) [310 mM sucrose, 10 mM Tris-HCl (pH 7.5) and 0.05% BSA (w/v), with protease inhibitor cocktail (Sigma)], homogenized with Dounce homogenizer, centrifuged at $1000 \times g$ for 10 min at 4 °C.

The supernatant was further centrifuged at $4500 \times g$ for 15 min at 4 °C, and the pellet was washed once with MIB. Crude mitochondrial pellets were resuspended in MIB with protease inhibitor cocktail. Further, isolated mitochondria were cross-linked with 1% paraformaldehyde and sonicated. Solubilized mitochondrial chromatin was immunoprecipitated with antibodies against HA, POLRMT and control IgG. Antibody–chromatin complexes were pulled down using protein A-sepharose (Millipore), washed, and then eluted. After crosslink reversal and proteinase K treatment, immunoprecipitated DNA was extracted with phenol-chloroform, and ethanol precipitated. The DNA fragments were further analyzed by qPCR. Primers for qPCR are listed in Dataset EV1.

## Chromatin immunoprecipitation and quantitative PCR (ChIP-qPCR)

ChIP-qPCR assays were performed as described previously (Wang et al, 2015). Briefly, cells were cross-linked with 1% paraformaldehyde and sonicated. Solubilized chromatin was immunoprecipitated with antibodies against E2F1 or negative control IgG. Antibody–chromatin complexes were pulled down using protein A-sepharose (Millipore), washed, and then eluted. After crosslink reversal and proteinase K treatment, immunoprecipitated DNA was extracted with phenol-chloroform, ethanol precipitated. The DNA fragments were further analyzed by qPCR. Primers for qPCR are listed in Dataset EV1.

## Mitochondria staining and mtDNA quantification

To quantify mitochondria mass, $1 \times 10^6$ cells were stained with 50 nM MitoTracker Green (Invitrogen) for 30 min at 37 °C and analyzed with a fluorescence microplate reader (BioTek, Ex. 491 nm, Em. 516 nm). To determine mtDNA copy number, total DNA was isolated using DNeasy kits (Qiagen). Samples were adjusted to 1 ng/μL (final concentration). Nuclear and mitochondrial DNA content was analyzed by qPCR with the method we previously used (Wang et al, 2021). In brief, mtDNA content was determined by normalizing mtDNA abundance [tRNA-Leu (UUR) in human and 16 S rRNA in mouse] to nDNA (beta-2-microglobulin, B2M) abundance. Primers for qPCR are listed in Dataset EV1.

## Generation of stable cell pools

pMKO.1-puro, pMKO.1-hygro, and pLKO.1-blast vectors were used for shRNA cloning. shRNAs targeting *SUCLG1*, *POLRMT*, *FLT3*, and *E2F1* were used to generate stable knockdown cell pools. Retrovirus was produced by using a two-plasmid packaging system (gag and vsvg). Cells were mixed with 8 μg/mL polybrene and spinfected with the retrovirus. Stable knockdown cells were achieved after being selected in puromycin (4 μg/mL), hygromycin (200 μg/mL), or blasticidin (5 μg/mL) or for 1 week.

To generate *SUCLG1*-knockdown and rescue stable cell pools, Flag-tagged wild-type *SUCLG1* or its mutants were cloned into the lentiviral pLV-EF1a-IRES-puro vector. HEK293T cells were used to produce lentiviruses. Cells stably expressing sh*SUCLG1*-#1 which targets the 3' UTR were used for transduction. After that, cells were selected in puromycin (4 μg/mL) for 1 week. To generate *POLRMT*-

knockdown and rescue stable cell pools, HA-tagged human wild-type *POLRMT* or its mutants containing two silent nucleotide substitutions in the sequence corresponding to the shRNA-targeted region were cloned into the lentiviral pLV-EF1a-IRES-puro vector. Lentivirus was produced and transduced into cells stably expressing sh*POLRMT*-#1. Cells were selected in puromycin (4 μg/mL) for 1 week.

## ETC complex activity

Mitochondria were purified from CD34$^+$ CB cells or leukemia cells using the Mitochondria Isolation Kit (Sigma # MITOISO2). Freshly isolated mitochondrial pellets were resuspended in PBS and loaded onto pre-coated microplate wells at a protein concentration of 0.2 mg/mL. Complex I (Abcam #ab109721), Complex II (Abcam #ab109908), Complex III (Abcam #ab287844), Complex IV (Abcam #ab109909), and Complex V (Cayman Chem #701000) activities were measured following the protocols provided by the manufacturer.

## ROS quantification

Reactive oxygen species (ROS) were monitored using CellROX green reagent (Invitrogen #C10444). Briefly, one million cells in 1000 μL of culture media were incubated with CellROX green at a final concentration of 5 μM. Cells were incubated for 30 min, and washed three times with PBS. ROS level was determined using flow cytometry (Ex. 488 nm; Em. 550 nm).

## BrdU cell proliferation assay

Cell proliferation was determined using BrdU Assay Kit (Cell signaling #6813). Briefly, cells were seeded at 500,000 cells/well. After adding BrdU, cells were cultured for 24 h and fixed for antibody-based detection of incorporated BrdU.

## Apoptosis, senescence, and cell cycle

Annexin V/7-AAD staining kit (Beyotime #C1062S) was used to evaluate apoptosis in leukemia cells. Cells were washed with Annexin binding buffer. Annexin V-FITC and 7-AAD were added and incubated for 15 min at 37 °C in the dark. Apoptotic cells were immediately analyzed by flow cytometry. At least 10,000 cells per sample were analyzed.

For doxorubicin-induced senescence, leukemia cells were treated with 0.1 μM doxorubicin for 4 days. β-Galactosidase Staining Kit (Beyotime #C0602) was used to detect senescence of control and *SUCLG1*-knockdown cells.

Cell cycle was analyzed by propidium staining kit to detect intracellular DNA content according to the manufacturer's instructions (Beyotime #C1052).

## Quantitative real-time PCR

Total RNA was isolated from cultured cells using the EasyPure RNA Purification Kit (Transgen) and reverse-transcribed with QuantScript RT Kit (Tiangen). The cDNA was preceded to real-time PCR with gene-specific primers in the presence of SYBR Green qPCR Master Mix (EZBioscience). PCR reactions were

performed in triplicate and the relative amount of cDNA was calculated by the comparative $C_T$ method using the β-actin as a control. Primer sequences are listed in Dataset EV1.

## Metabolite quantification

To quantify intracellular metabolites, $5 \times 10^6$ cells were washed in 1 mL saline (0.9% NaCl in water). To quantify mitochondrial succinyl-CoA, mitochondria were isolated from $1 \times 10^7$ cells and washed in 1 mL saline (0.9% NaCl in water). Further, samples were lysed in ice-cold methanol (HPLC grade). Water and chloroform were added to the lysate (methanol:water:chloroform=2:1:2 final volume), followed by vortexing and centrifugation ($5000 \times g$, 4 °C for 15 min). The aqueous phase was dried with nitrogen flow evaporator at 37 °C. Extracted metabolites were resuspended in 50% acetonitrile and subjected to mass spectrometry analysis. Relative metabolite abundance was calculated by normalizing to total cellular or mitochondrial protein as indicated.

## Oxygen consumption rate (OCR) and extracellular acidification rate (ECAR)

OCR was determined using the XFe96 Extracellular Flux Analyzer (Agilent). Briefly, leukemia cells were attached to 96-well plates using Cell-Tak (Corning) at the density of $4 \times 10^4$ or $8 \times 10^4$ cells/well, respectively. Cells were incubated with Seahorse XF RPMI medium buffer (without phenol red, with 10 mM glucose, 2 mM glutamine, and 1 mM pyruvate). Cell Mito Stress Test Kit (Agilent) was used to measure cellular mitochondrial function, 180 μL of Seahorse buffer plus 20 μL each of 2 μM oligomycin, 2 μM FCCP, and 0.5 μM rotenone/antimycin A (AA) was automatically injected to determine the oxygen consumption rate (OCR). The Glycolysis Stress Test Kit (Agilent) was used to measure the glycolytic capacity, 25 μL each of 10 mM glucose, 2 μM oligomycin, and 100 mM 2-deoxyglucose (2DG) were added to determine the ECAR. OCR and ECAR values were normalized to cell numbers.

## Glucose uptake, glutamine consumption, and lactate production

Leukemia cells were resuspended in fresh media and seeded to six-well plates at a density of 3 million cells/mL. After incubation for 6 h, cells were removed by centrifugation. The supernatants and fresh media were deproteinized using 10 K spin column (Abcam). Lactate production (Sigma #MAK064), glucose uptake (Sigma #MAK263), and glutamine consumption (Abcam, # ab197011) were determined by fluorescence/colorimetric-based assay kits following the manufacturers' instructions and normalized to cell number or total protein as indicated.

## Metabolic enzyme activity assay

SUCLG1 enzyme activity was assayed by measuring CoA-SH accumulation as previously described (Alarcon et al, 2002). Briefly, Flag-tagged SUCLG1 proteins were overexpressed in cells, immunoprecipitated with Flag beads, eluted by Flag peptides (GL Biochem), and subjected to activity assay with succinyl-CoA and ATP as substrates. The reaction mixture consists of 10 mM Tris-HCl (pH 7.4), 10 mM MgCl₂, 0.2 mM 5,5-dithio-bis-(2-nitrobenzoic acid) (DTNB), 1 mM succinyl-CoA and 0.1 mM ATP in a total volume of 200 μL. Reactions were initiated by adding an immunopurified enzyme, which released CoA-SH from succinyl-CoA. Free CoA-SH further reacted with DTNB to generate colored product. After incubation for 15 min at 25 °C, the accumulation of CoA-SH was monitored spectrophotometrically with a microplate reader (BioTek, 415 nm). A control reaction without SUCLG1 enzyme was included to determine nonspecific hydrolysis of CoA derivatives. Enzyme activity was normalized to SUCLG1 protein.

SDH activity in whole-cell lysate was determined by generating a product with absorbance at 600 nm (Sigma #MAK197). FH activity was determined by immobilizing FH enzyme onto microplates. FH activity was coupled to the reaction catalyzed by malate dehydrogenase, following the procedures provided by the manufacturer (Abcam #ab110043). Enzyme activities are normalized to total protein. Chemical inhibitors against SDH (SDH-IN-1, 6 μM, MedChemExpress # HY-139983) and FH (Fumarate hydratase-IN-1, 3 μM, MedChemExpress #HY-100004) were added to the reaction mixture to validate the specificity of enzyme activity assay.

## In vitro succinylation assay

Bacterially purified His-tagged POLRMT was incubated with increasing concentrations of succinyl-CoA or succinate in the presence or absence of immunopurified SUCLG-Flag. The reaction buffer for in vitro succinylation consists 50 mM Tris-HCl (pH 7.4) and 150 mM NaCl. The mixture was incubated at 37 °C for 60 min. Products were subjected to western blotting to detect succinylation.

## In organello transcription

Mitochondria were isolated from MV411 cells using the Mitochondria Isolation Kit following the protocol provided by the manufacturer (Sigma). For in organello transcription experiments, freshly purified mitochondria (800 μg) were washed in incubation buffer (25 mM sucrose, 75 mM sorbitol, 10 mM Tris-HCl, 10 mM K₂HPO4, 100 mM KCl, 0.05 mM EDTA, 1 mM ATP, 5 mM MgCl₂, 10 mM glutamate, 2.5 mM malate, and 1 mg/mL BSA, pH 7.4) for three times and resuspended in 750 μL of incubation buffer. The mixture was further supplemented with 50 μCi of α-³²P-UTP (PerkinElmer). Samples were incubated at 37 °C for an hour. Afterward, mitochondria were pelleted, resuspended in incubation buffer containing 0.2 mM UTP, and incubated for 10 min at 37 °C. Mitochondria were subsequently washed twice in ice-cold wash buffer (10% glycerol, 0.15 mM MgCl₂, and 10 mM Tris-HCl, pH 6.8). RNA was extracted using TRIzol (Invitrogen) and analyzed by northern blotting. The membrane was exposed to a phosphor-imager screen. POLRMT inhibitor (IMT1, 2.5 μM) was added to resuspended mitochondria and incubated for 20 min before supplementing radiolabeled UTP. POLRMTi-treated samples were included as a negative control for detecting mitochondrial transcription activity. An aliquot of mitochondria (20 μL) was collected for detecting the loading control HSP60.

## Cell proliferation and colony formation assay

In the cell proliferation assay, cells ($1 \times 10^5$/mL) were seeded into six-well plates. Viable cells were visualized by methylene blue

staining and counted every day for 4 days. In colony formation assay, AML cells were plated in methylcellulose medium (MethoCult H4434; Stem Cell Technologies). In total, 2000 cells in 0.5 mL IMDM with 10% FBS were added to 3.5 mL of methylcellulose medium. After thorough vortex mixing, the cell suspension was plated into the six-well plate with 1 mL in each well. Culture plates were incubated at 37 °C in a humid atmosphere with 5% $CO_2$. Colonies (>50 mm diameter) were counted after 7 days of incubation.

### Recombinant protein

One-shot *E. coli* BL21 (DE3) (Invitrogen) was grown in LB medium at 37 °C and then at 16 °C after IPTG induction, for recombinant POLRMT-His expression. Ni-NTA magnetic agarose beads (Qiagen # 36113) were used for immobilizing and purifying POLRMT-His protein.

### Mouse leukemia models

All animal procedures were performed following the approval (# 2023-A0012-02) of the animal care committee at Shanghai General Hospital, Shanghai Jiao Tong University School of Medicine and compliance with ethical regulations. Black 6 (B6) mice (C57BL/6J) and NSG mice (NOD *scid* gamma) were purchased from the Jackson Laboratory and housed in 12/12 LD with ad libitum feeding. To establish the mouse leukemia model, retroviral vector pMIG-FLAG-MLL-AF9 (GFP as selection marker), lentiviral vector pLV-puro-FLT3ITD-HA, and pMKO.1-hygro-sh*Suclg1* were used for virus packaging. Two days after transduction, cells were selected with puromycin (4 μg/mL) and hygromycin (200 μg/mL) for 7 days. GFP-positive cells were flow sorted and $1 \times 10^5$ cells were transplanted into sublethally irradiated recipient mice (6 Gy, 6-week-old male). To establish NSG model of human leukemia xenograft, stable leukemia cells were isolated during the exponential growth phase from in vitro cell culture. One million cells were transplanted into NSG mice 24 h after sublethal irradiation (2 Gy, 8-week-old male). FLT3i was prepared in 40% PEG30, 5% Tween-80, and 45% PBS. Treatment was started 2 days after transplantation, FLT3i was intraperitoneally injected at 10 mg/kg every 48 h.

### Human bone marrow samples and *FLT3* mutation detection

Normal and leukemic human bone marrow samples were collected from diagnostic bone marrow aspirations at Southwest Hospital. Patient ages at diagnosis ranged from 35 to 72 years old (7 males and 8 females, Chinese) in this study. Bone marrow mononuclear cells were isolated by density gradient centrifugation and stored in liquid nitrogen until further use. Written informed consent was obtained from all subjects and that the experiments conformed to the principles set out in the WMA Declaration of Helsinki and the Department of Health and Human Services Belmont Report. The procedures related to human subjects were approved (#2023SQ021) by the Institutional Ethics Review Board of Southwest Hospital and Shanghai General Hospital, Shanghai Jiao Tong University School of Medicine. The mutation status of the TKD and ITD of *FLT3* was determined as previously described (Schumacher et al, 2020). Briefly, genomic DNA was extracted from $5 \times 10^6$ cells. Exons 14 to

15 (ITD) and exon 20 (TKD) were PCR amplified with primers listed in Dataset EV1. PCR products were digested with EcoRV-HF (New England Biolabs), mixed with Hi-Di formamide (Thermo-Fisher Scientific) and MapMarker DNA sizing standards (BioVentures), and further resolved on a DNA analyzer (ABI 3730). In TKD reaction, the signal ratio was determined as mutated peak area divided by wild-type peak area. In ITD reaction, the signal ratio was presented as the sum of all mutated peak areas divided by the wild-type peak area. Samples with signal ratios >0.25 were adopted for analysis in this study.

### Quantification and statistical analysis

Relative concentrations of immunoprecipitated proteins were quantified by measuring the band intensity of western blots by using ImageQuant TL (Amersham Biosciences). Consensus sequences of differentially expressed mitochondrial genes were visualized with the R package ggseqlogo (Wagih, 2017). Student's *t* test or one-way ANOVA with Dunnett's multiple comparisons test was performed to determine statistical significance using GraphPad Prism. Leukemia survival was plotted on Kaplan–Meier survival curve and analyzed with log-rank (Mantel–Cox) test. All data shown represent the results obtained from at least three independent experiments. The *P* values < 0.05 were considered statistically significant. Experimenters were not blinded to group assignment and outcome assessment.

## Data availability

This study includes no data deposited in external repositories. Spearman's correlations of mitochondrial metabolic genes with ETC genes are available from cBioPortal (https://www.cbioportal.org/). GSE237299, GSE227839, GSE185824 and GSE226461 are available from GEO dataset (https://www.ncbi.nlm.nih.gov/): https://www.ncbi.nlm.nih.gov/search/all/?term=GSE237299, https://www.ncbi.nlm.nih.gov/search/all/?term=GSE227839, https://www.ncbi.nlm.nih.gov/search/all/?term=GSE185824, https://www.ncbi.nlm.nih.gov/search/all/?term=GSE226461.

The source data of this paper are collected in the following database record: biostudies:S-SCDT-10_1038-S44318-024-00101-9.

## Peer review information

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

## Acknowledgements

Y-PW is supported by the National Key Research and Development Program of China (No. 2022YFA0807100), National Natural Science Foundation of China (No. 32322024), the Natural Science Foundation of Shanghai (No. 22ZR1414200), and the Shanghai Rising-Star Program (No. 20QA1401700). SZ is supported by the National Natural Science Foundation of China (No.

32100923). We thank the Core Facility of Basic Medical Sciences (Shanghai Jiao Tong University School of Medicine) for technical support.

## Author contributions

**Weiwei Yan**: Data curation; Formal analysis. **Chengmei Xie**: Data curation; Formal analysis; Methodology; Writing—original draft; Writing—review and editing. **Sijun Sun**: Data curation; Software; Formal analysis; Visualization; Methodology; Writing—original draft; Writing—review and editing. **Quan Zheng**: Data curation. **Jingyi Wang**: Data curation; Formal analysis; Methodology. **Zihao Wang**: Data curation. **Cheuk-Him Man**: Formal analysis; Methodology. **Haiyan Wang**: Data curation; Methodology. **Yunfan Yang**: Data curation; Methodology; Writing—review and editing. **Tianshi Wang**: Data curation; Formal analysis; Methodology. **Leilei Shi**: Data curation; Methodology. **Shengjie Zhang**: Data curation; Formal analysis; Supervision; Methodology; Writing—original draft; Writing—review and editing. **Chen Huang**: Data curation. **Shuangnian Xu**: Data curation; Formal analysis; Investigation; Methodology; Writing—original draft; Writing—review and editing. **Yi-Ping Wang**: Conceptualization; Formal analysis; Supervision; Funding acquisition; Investigation; Writing—original draft; Project administration; Writing—review and editing.

Source data underlying figure panels in this paper may have individual authorship assigned. Where available, figure panel/source data authorship is listed in the following database record: biostudies:S-SCDT-10_1038-S44318-024-00101-9.

## Disclosure and competing interests statement

The authors declare no competing interests.

# Expanded View Figures

**Figure EV1. SUCLG1 maintains mtDNA-encoded gene expression and mitochondrial mass.**

(**A**) Schematic overview of TCA cycle and the reaction catalyzed by SUCLG1. (**B**) Gene set enrichment analysis of the relation between *SUCLG1* and ETC, also known as oxidative phosphorylation, in TCGA AML and two other AML transcriptomic studies (GSE227839 and GSE185824), Kolmogorov–Smirnov test. (**C**) The top ten mitochondrial metabolic genes that positively correlate with ETC genes in TCGA AML. Shown are Spearman's correlation coefficients. (**D**) Scrambled control or shRNAs against *SUCLG1* were transduced into human CD34$^+$ CB cells or cancer cell lines as indicated. Total DNA was extracted to quantify mtDNA. Data represent means ± SD, $n = 3$ independent biological replicates, *t* test. **$P < 0.01$. (**E**) Mitochondria were isolated from control or *SUCLG1*-knockdown MV411 cells. Whole-cell lysate and mitochondrial fractions were subject to western blotting. Histone H3, G6PD and HSP60 serve as markers for nucleus, cytosol and mitochondria, respectively. (**F**) Shown is the correlation between *SUCLG1* and ETC-related genes in various TCGA cancers. (**G, H**) mRNA expression of ETC-correlated genes was determined in control and *SUCLG1*-knockdown MV411 or HL60 cells. Data represent means ± SD, $n = 3$ independent biological replicates, *t* test. **$P < 0.01$; *$P < 0.05$; n.s. not significant. (**I, J**) Control and *SUCLG1*-knockdown MV411 cells were stained with CellROX Green (**I**). Cellular ROS levels were determined (**J**). Data represent means ± SD, $n = 3$ independent biological replicates, *t* test. **$P < 0.01$. (**K, L**) Apoptosis of control and *SUCLG1*-knockdown MV411 cells was determined. Data represent means ± SD, $n = 3$ independent biological replicates, *t* test. **$P < 0.01$. (**M**) Stable MV411 cells were treated with 0.1 μM doxorubicin for 4 days, senescence was visualized by β-galactosidase staining. Scale bars: 50 μm. (**N**) BrdU incorporation was assayed in control and *SUCLG1*-knockdown MV411 cells. Data represent means ± SD, $n = 3$ independent biological replicates, *t* test. **$P < 0.01$. (**O**) Distribution of stable MV411 cells in cell cycle was determined. Data represent means ± SD, $n = 3$ independent biological replicates, *t* test. **$P < 0.01$; *$P < 0.05$. (**P**) Multiple alignments of amino acid sequences adjacent to P170 and A209 of SUCLG1. (**Q, R**) The structural locations of P170 and A209 residues were analyzed (PDB ID: 6XRU). The catalytic center contains succinate, Mg$^{2+}$, and desulfo-CoA (succinyl-CoA analog). (**S**) Flag-tagged SUCLG1 and disease-derived mutants (P170R and A209E) were expressed in HEK293T cells. SUCLG1 enzymes were immunopurified and subjected to catalytic activity assay. Enzyme activities were normalized to Flag-tagged protein. Data represent means ± SD, $n = 3$ independent biological replicates, *t* test. **$P < 0.01$. Source data are available online for this figure.

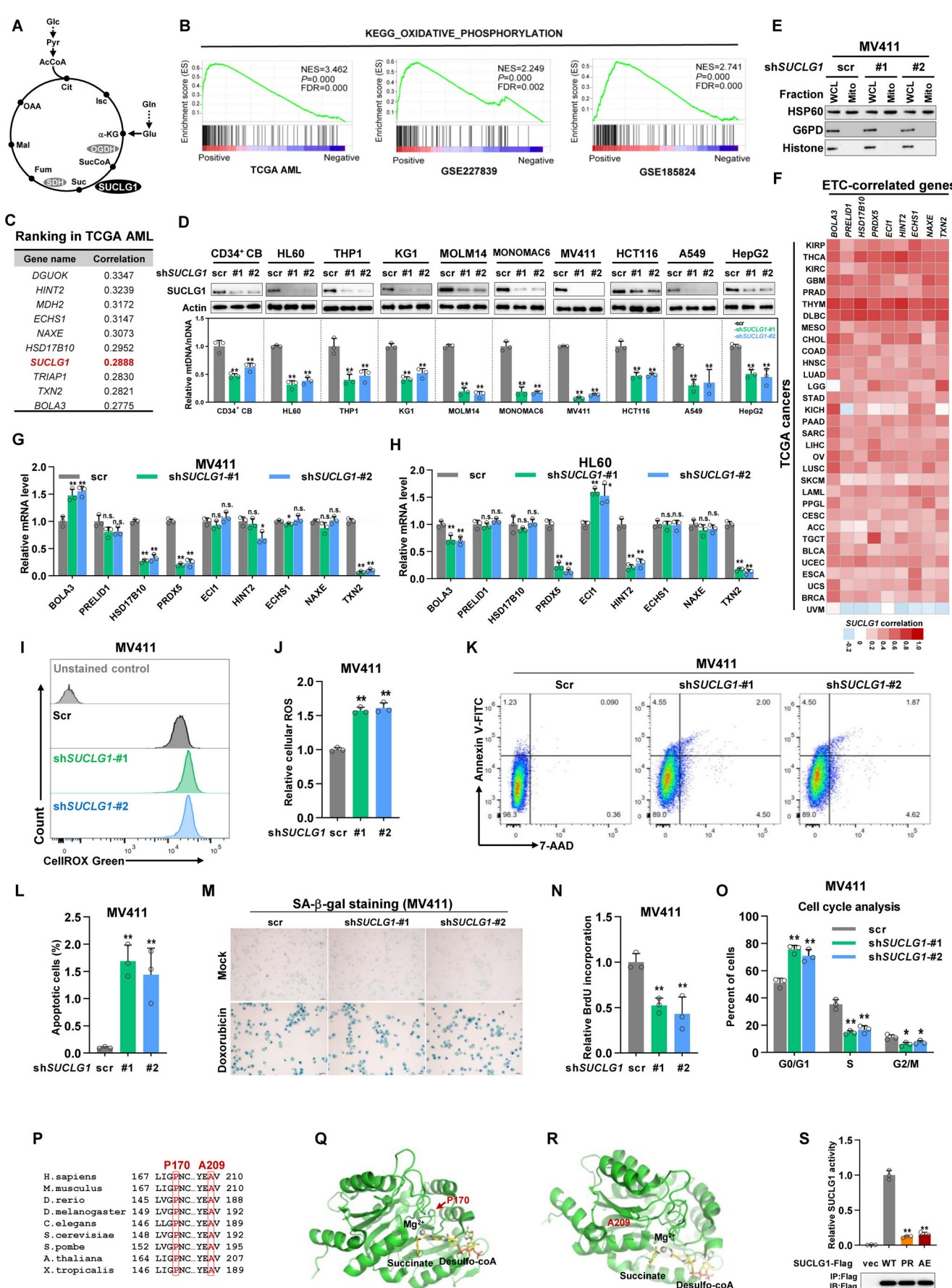

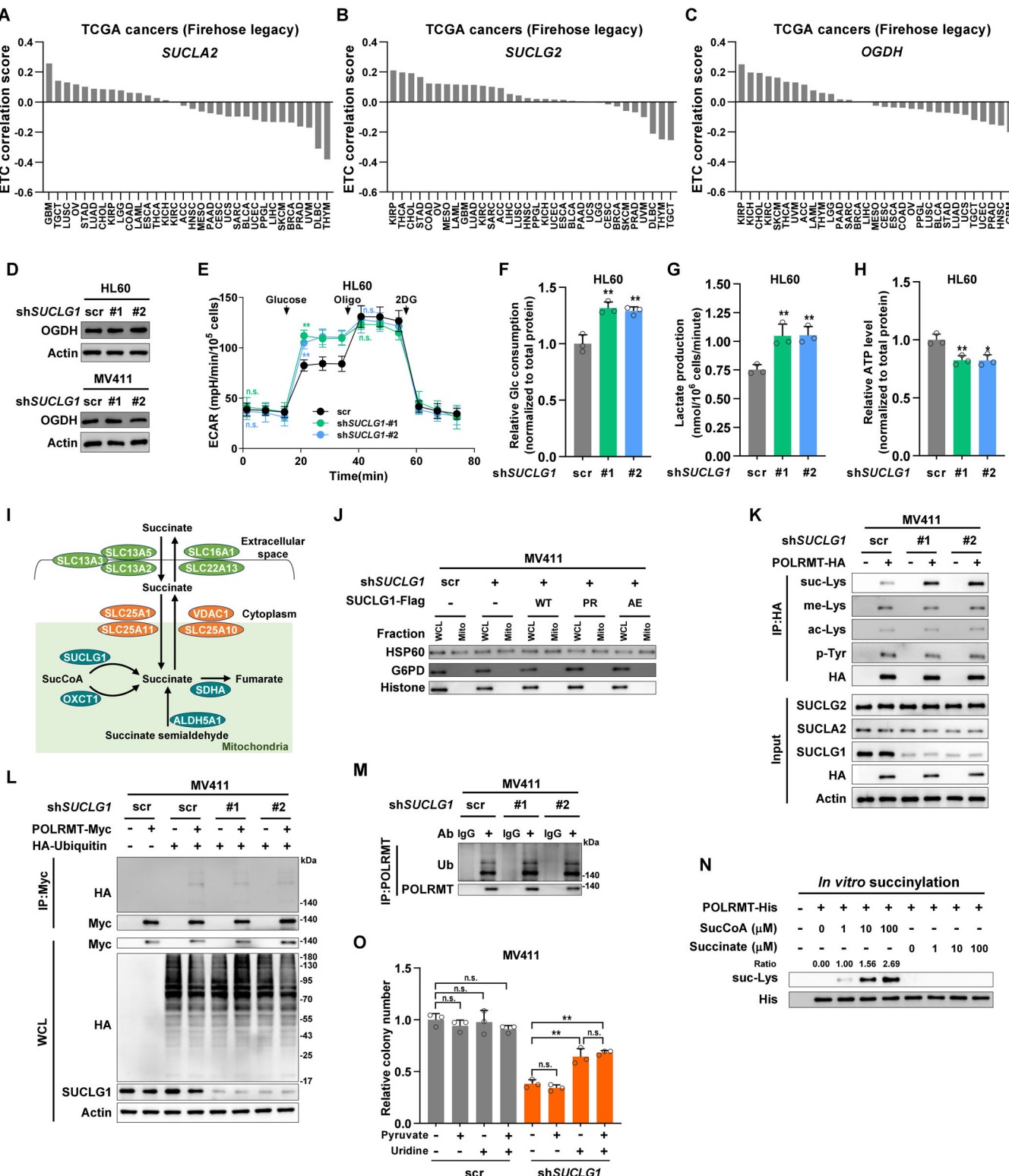

**Figure EV2. SUCLG1 reduces succinyl-CoA level to restrict POLRMT succinylation.**

(A–C) ETC correlation scores of succinyl-CoA-metabolizing enzymes, *SUCLA2* (A), *SUCLG2* (B), and *OGDH* (C), were determined in various TCGA cancers. Shown are Spearman's correlation coefficients. (D) OGDH protein expression in stable HL60 and MV411 cells was determined by western blotting. (E) Extracellular acidification rates of control or *SUCLG1*-knockdown HL60 cells were determined. Data represent means ± SD, $n = 3$ independent biological replicates, *t* test. **$P < 0.01$; n.s. not significant. (F, G) Glucose (Glc) consumption and lactate production were assayed in control and *SUCLG1*-knockdown HL60 cells. Data represent means ± SD, $n = 3$ independent biological replicates, *t* test. **$P < 0.01$. (H) ATP abundance was quantified in control and *SUCLG1*-knockdown HL60 cells, and normalized to total protein. Data represent means ± SD, $n = 3$ independent biological replicates, *t* test. **$P < 0.01$; *$P < 0.05$. (I) Schematic overview of metabolic enzymes and transporters involved in succinate metabolism. (J) Mitochondria were isolated from stable *SUCLG1*-knockdown and rescue MV411 cells. Whole-cell lysate and mitochondrial fractions were subject to western blotting. Histone H3, G6PD, and HSP60 were included as markers for the nucleus, cytosol, and mitochondria, respectively. (K) HA-tagged POLRMT was expressed in scrambled control or *SUCLG1*-knockdown MV411 cells. POLRMT-HA was immunoprecipitated from stable MV411 cells and subjected to western blotting to determine lysine succinylation (suc-Lys), lysine methylation (me-Lys), lysine acetylation (ac-Lys), and tyrosine phosphorylation (p-Tyr). (L) Myc-tagged POLRMT and HA-tagged ubiquitin were co-expressed in stable MV411 cells. POLRMT was immunopurified with Myc antibody to determine its ubiquitination by western blotting. (M) Endogenous POLRMT was immunoprecipitated from control or *SUCLG1*-knockdown MV411 cells. Ubiquitination was determined by western blotting. (N) Bacterially expressed His-tagged POLRMT was purified and incubated with succinyl-CoA (sucCoA) or succinate in vitro. Lysine succinylation levels were determined by western blotting and normalized to POLRMT-His protein (ratio). (O) Scrambled control or *SUCLG1*-knockdown (sh-#1) MV411 cells were cultured with 1 mM pyruvate or 0.2 mM uridine. Colony formation was determined. Data represent means ± SD, $n = 3$ independent biological replicates, *t* test. **$P < 0.01$; n.s. not significant. Source data are available online for this figure.

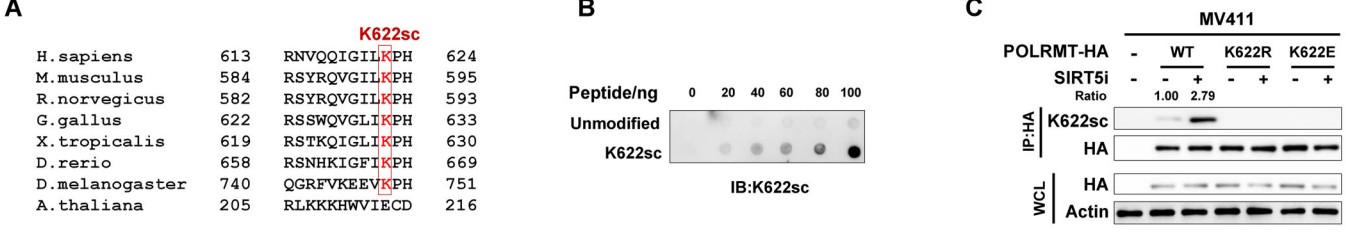

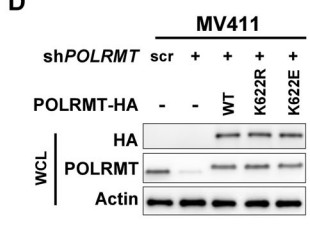

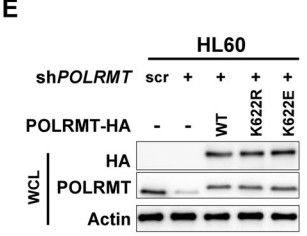

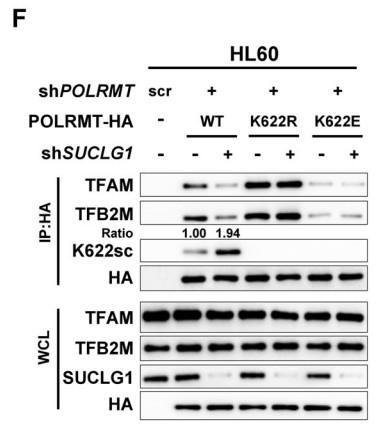

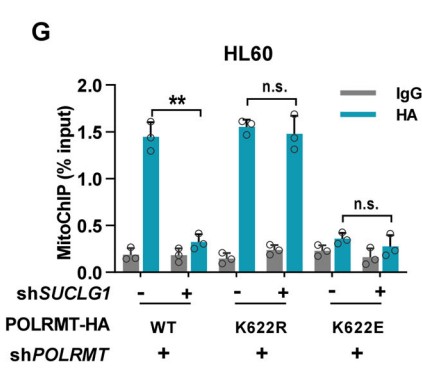

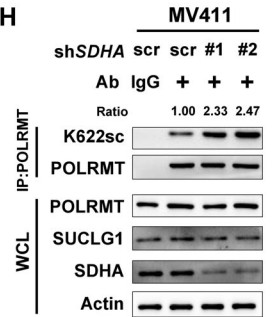

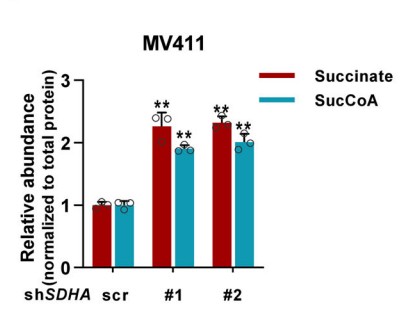

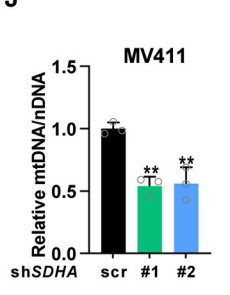

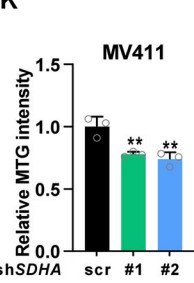

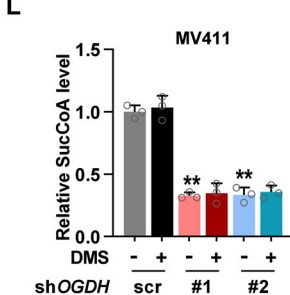

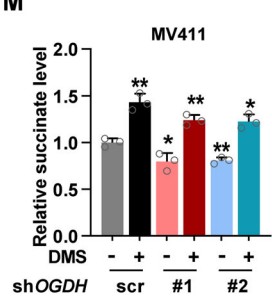

◀ **Figure EV3. K622 succinylation suppresses POLRMT activity.**

(A) Multiple alignments of amino acid sequences corresponding to K622 of POLRMT from various model organisms. (B) Dot blotting was performed to test the specificity of K622 succinylation antibody. Unmodified peptide was included as the negative control. (C) HA-tagged POLRMT and its K622 mutants were expressed in MV411 cells. Cells were treated with SIRT5 inhibitor (SIRT5i). POLRMT was immunopurified with HA antibody to determine K622 succinylation. K622sc signal was normalized to POLRMT-HA protein (ratio). (D, E) HA-tagged POLRMT and its K622 mutants were re-expressed in *POLRMT*-knockdown MV411 and HL60 cells at physiologically relevant levels. (F, G) Control or shRNA against *SUCLG1* was expressed in *POLRMT*-knockdown and rescue HL60 cells. POLRMT-HA was immunopurified to assay protein interaction and K622 succinylation. K622sc signal was normalized to POLRMT-HA protein (ratio) (F). MitoChIP was performed to determine mtDNA binding of POLRMT-HA (G). Data represent means ± SD, $n = 3$ independent biological replicates, $t$ test. **$P < 0.01$; n.s. not significant. (H) Endogenous POLRMT was immunoprecipitated from control or *SDHA*-knockdown MV411 cells. K622 succinylation was determined by western blotting. (I) Metabolites were extracted from control and *SDHA*-knockdown MV411 cells. Succinate and succinyl-CoA levels were quantified by mass spectrometry and normalized to total protein. Data represent means ± SD, $n = 3$ independent biological replicates, $t$ test. **$P < 0.01$. (J, K) mtDNA abundance and mitochondrial mass were determined by qPCR and MTG staining, respectively. Data represent means ± SD, $n = 3$ independent biological replicates, $t$ test. **$P < 0.01$. (L, M) Scrambled control or *OGDH*-knockdown stable MV411 cells were cultured with or without 50 μM dimethyl succinate (DMS). Succinyl-CoA and succinate levels were quantified by mass spectrometry and normalized to total protein. Data represent means ± SD, $n = 3$ independent biological replicates, $t$ test. **$P < 0.01$; *$P < 0.05$. Source data are available online for this figure.

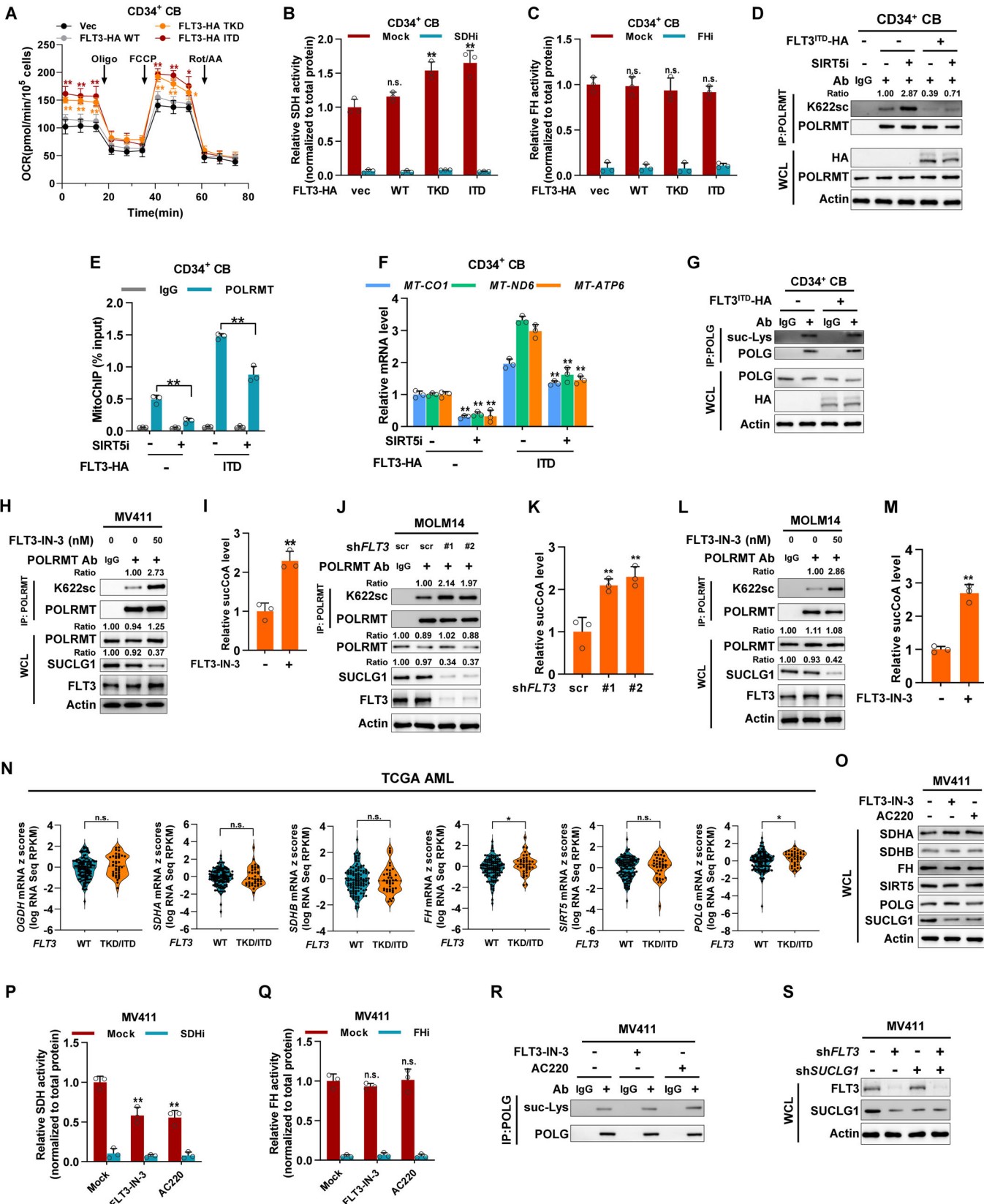

◄ **Figure EV4.** **Leukemia-derived FLT3 mutants upregulate SUCLG1 to boost mitobiogenesis.**

(A) HA-tagged FLT3 and its mutants were stably expressed in CD34$^+$ CB cells. Oxygen consumption rates were determined by Seahorse experiments. Data represent means ± SD, $n = 3$ independent biological replicates, $t$ test. **$P < 0.01$; *$P < 0.05$. (B, C) Enzymatic activities of SDH and FH in stable CD34$^+$ CB cells were determined and normalized to total protein. Data represent means ± SD, $n = 3$ independent biological replicates, $t$ test. **$P < 0.01$; n.s. not significant. (D–F) Stable CD34$^+$ CB cells were treated with or without SIRT5 inhibitor. K622 succinylation (D), mtDNA binding (E), and mtDNA-encoded gene expression (F) were determined. Data represent means ± SD, $n = 3$ independent biological replicates, $t$ test. **$P < 0.01$. (G) Endogenous POLG was immunoprecipitated from control or FLT3$^{ITD}$-expressing CD34$^+$ CB cells. Lysine succinylation was determined by western blotting. (H, I) MV411 cells were treated with FLT3-IN-3, a chemical inhibitor of FLT3, for 24 h. Endogenous POLRMT was immunopurified to determine K622 succinylation. K622sc levels were normalized to POLRMT protein (ratio) (H). Cellular succinyl-CoA levels were determined and normalized to total protein (I). Data represent means ± SD, $n = 3$ independent biological replicates, $t$ test. **$P < 0.01$. (J, K) Endogenous POLRMT was immunopurified from scrambled control or FLT3-knockdown MOLM14 cells to determine K622 succinylation. K622sc levels were normalized to POLRMT protein (ratio) (J). Cellular succinyl-CoA levels were determined and normalized to total protein (K). Data represent means ± SD, $n = 3$ independent biological replicates, $t$ test. **$P < 0.01$. (L, M) MOLM14 cells were treated with FLT3-IN-3 for 24 h. Endogenous POLRMT was immunopurified to determine K622 succinylation. K622sc levels were normalized to POLRMT protein (ratio) (L). Cellular succinyl-CoA levels were determined and normalized to total protein (M). Data represent means ± SD, $n = 3$ independent biological replicates, $t$ test. **$P < 0.01$. (N) mRNA expression of mitochondrial genes in TCGA AML dataset as indicated was extracted from cBioportal. Gene expression levels in FLT3$^{WT}$ ($n = 116$) and FLT3$^{TKD/ITD}$ ($n = 39$) samples were compared. $t$ test. *$P < 0.05$. (O–R) MV411 cells were treated with FLT3 inhibitors. Protein expression of indicated genes was determined by western blotting (O). Enzyme activities of SDH (P) and FH (Q) were assayed and normalized to total protein. Endogenous POLG was immunoprecipitated to evaluate lysine succinylation (R). Data represent means ± SD, $n = 3$ independent biological replicates, $t$ test. **$P < 0.01$; n.s. not significant. (S) Scrambled control or shRNAs targeting FLT3 (#1) and SUCLG1 (#1) were co-expressed in MV411 cells as indicated. Knockdown efficiency was tested by western blotting. Source data are available online for this figure.

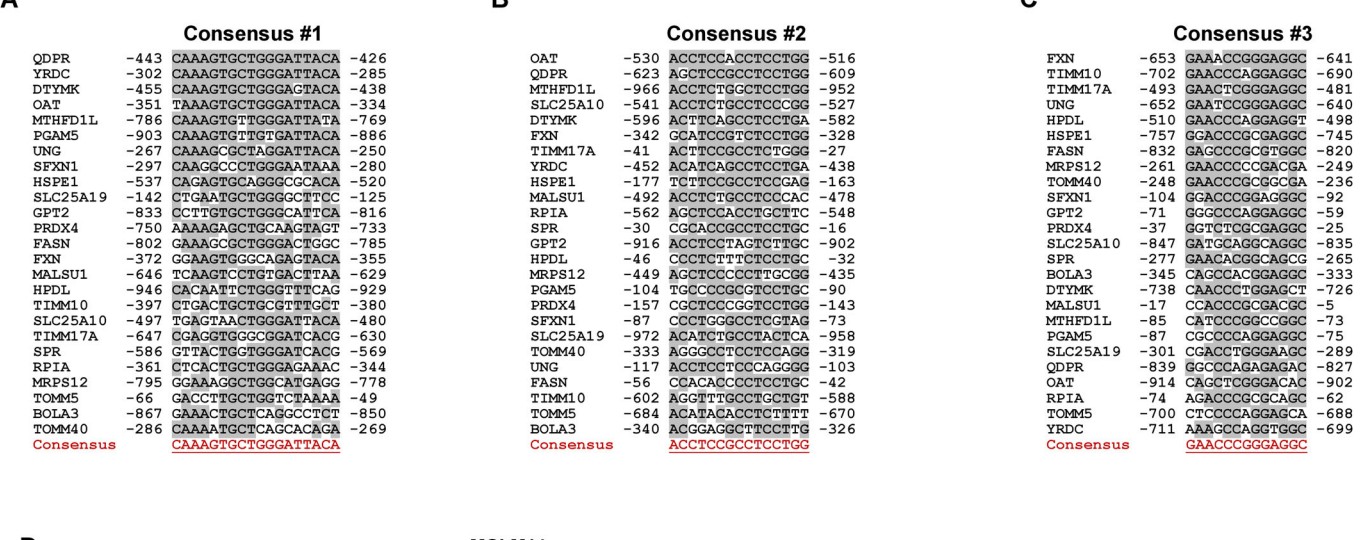

**A**

### Consensus #1

| | | | |
|---|---|---|---|
| QDPR | -443 | CAAAGTGCTGGGATTACA | -426 |
| YRDC | -302 | CAAAGTGCTGGGATTACA | -285 |
| DTYMK | -455 | CAAAGTGCTGGGAGTACA | -438 |
| OAT | -351 | TAAAGTGCTGGGGATTACA | -334 |
| MTHFD1L | -786 | CAAAGTGTTGGGATTATA | -769 |
| PGAM5 | -903 | CAAAGTGTTGTGATTACA | -886 |
| UNG | -267 | CAAAGCGCTAGGATTACA | -250 |
| SFXN1 | -297 | CAAGGCCCTGGGAATAAA | -280 |
| HSPE1 | -537 | CAGAGTGCAGGGCGCACA | -520 |
| SLC25A19 | -142 | CTGAATGCTGGGGCTTCC | -125 |
| GPT2 | -833 | CCTTGTGCTGGGCATTCA | -816 |
| PRDX4 | -750 | AAAAGAGCTGCAAGTAGT | -733 |
| FASN | -802 | GAAACGCTGGGACTGGC | -785 |
| FXN | -372 | GGAAGTGGGCAGAGTACA | -355 |
| MALSU1 | -646 | TCAAGTCCTGTGACTTAA | -629 |
| HPDL | -946 | CACAATTCTGGGTTTCAG | -929 |
| TIMM10 | -397 | CTGACTGCTGCGTTTGCT | -380 |
| SLC25A10 | -497 | TGAGTAACTGGGATTACA | -480 |
| TIMM17A | -647 | CGAGGTGGGCGGATCACG | -630 |
| SPR | -586 | GTTACTGGTGGGATCACG | -569 |
| RPIA | -361 | CTCACTGCTGGGAGAAAC | -344 |
| MRPS12 | -795 | GGAAAGGCTGGCATGAGG | -778 |
| TOMM5 | -66 | GACCTTGCTGGTCTAAAA | -49 |
| BOLA3 | -867 | GAAACTGCTCAGGCCTCT | -850 |
| TOMM40 | -286 | CAAAATGCTCAGCACAGA | -269 |
| **Consensus** | | **CAAAGTGCTGGGATTACA** | |

**B**

### Consensus #2

| | | | |
|---|---|---|---|
| OAT | -530 | ACCTCCACCTCCTGG | -516 |
| QDPR | -623 | AGCTCCGCCTCCTGG | -609 |
| MTHFD1L | -966 | ACCTCTGGCTCCTGG | -952 |
| SLC25A10 | -541 | ACCTCTGCCTCCCGG | -527 |
| DTYMK | -596 | ACTTCAGCCTCCTGA | -582 |
| FXN | -342 | GCATCCGTCTCCTGG | -328 |
| TIMM17A | -41 | ACTTCCGCCTCTGGG | -27 |
| YRDC | -452 | ACATCAGCCTCCTGA | -438 |
| HSPE1 | -177 | TCTTCCGCCTCCGAG | -163 |
| MALSU1 | -492 | ACCTCTGCCTCCCAC | -478 |
| RPIA | -562 | AGCTCCACCTGCTTC | -548 |
| SPR | -30 | CGCACCGCCTCCTGC | -16 |
| GPT2 | -916 | ACCTCCTAGTCTTGC | -902 |
| HPDL | -46 | CCCTCTTTCTCCTGC | -32 |
| MRPS12 | -449 | AGCTCCCCCTTGCGG | -435 |
| PGAM5 | -104 | TGCCCCGCGTCCTGC | -90 |
| PRDX4 | -157 | CGCTCCCGGTCCTGG | -143 |
| SFXN1 | -87 | CCCTGGGCGTCGTAG | -73 |
| SLC25A19 | -972 | ACATCTGCCTACTCA | -958 |
| TOMM40 | -333 | AGGGCCTCCTCCAGG | -319 |
| UNG | -117 | ACCTCCTCCCAGGGG | -103 |
| FASN | -56 | CCACACCCCTCCTGC | -42 |
| TIMM10 | -602 | AGGTTTGCCTGCTGT | -588 |
| TOMM5 | -684 | ACATACACCTCTTTT | -670 |
| BOLA3 | -340 | ACGGAGGCTTCCTTG | -326 |
| **Consensus** | | **ACCTCCGCCTCCTGG** | |

**C**

### Consensus #3

| | | | |
|---|---|---|---|
| FXN | -653 | GAAACCGGGAGGC | -641 |
| TIMM10 | -702 | GAACCCAGGAGGC | -690 |
| TIMM17A | -493 | GAACTCGGGAGGC | -481 |
| UNG | -652 | GAATCCGGGGAGGC | -640 |
| HPDL | -510 | GAACCCAGGAGGT | -498 |
| HSPE1 | -757 | GGACCCGCGAGGC | -745 |
| FASN | -832 | GAGCCCGCGTGGC | -820 |
| MRPS12 | -261 | GAACCCCCGACGA | -249 |
| TOMM40 | -248 | GAACCCGCGGCGA | -236 |
| SFXN1 | -104 | GGACCCGGAGGGC | -92 |
| GPT2 | -71 | GGGCCCAGGAGGC | -59 |
| PRDX4 | -37 | GGTCTCGCGAGGC | -25 |
| SLC25A10 | -847 | GATGCAGGCAGGC | -835 |
| SPR | -277 | GAACACGGCAGCG | -265 |
| BOLA3 | -345 | CAGCCACGGAGGC | -333 |
| DTYMK | -738 | CAACCCTGGAGCT | -726 |
| MALSU1 | -17 | CCACCCGCGACGC | -5 |
| MTHFD1L | -85 | CATCCCGGCCGGC | -73 |
| PGAM5 | -87 | CGCCCCAGGAGGC | -75 |
| SLC25A19 | -301 | CGACCTGGGAAGC | -289 |
| QDPR | -839 | GGCCCAGAGAGAC | -827 |
| OAT | -914 | CAGCTCGGGACAC | -902 |
| RPIA | -74 | AGACCCGCGCAGC | -62 |
| TOMM5 | -700 | CTCCCCAGGAGCA | -688 |
| YRDC | -711 | AAAGCCAGGTGGC | -699 |
| **Consensus** | | **GAACCCGGGAGGC** | |

**D**

WCL

MOLM14

shE2F1   scr   #1   #2

E2F1

Actin

**E**

MOLM14

Relative *SUCLG1* mRNA

shE2F1   scr   #1   #2

** **P < 0.01

---

**Figure EV5.  FLT3 signaling modulates nuclear transcription to enhance mitochondrial respiration.**

(A–C) Multiple alignments of consensus sequences in FLT3-regulated mitochondrial genes. Numbers indicate the distance (bp) to TSS. (D, E) Scrambled control or shRNAs targeting *E2F1* were expressed in MOLM14 cells. The knockdown efficiency was validated with western blotting (D). Total RNA was extracted to determine the mRNA expression of *SUCLG1* (E). Data represent means ± SD, $n = 3$ independent biological replicates, $t$ test. **$P < 0.01$. Source data are available online for this figure.

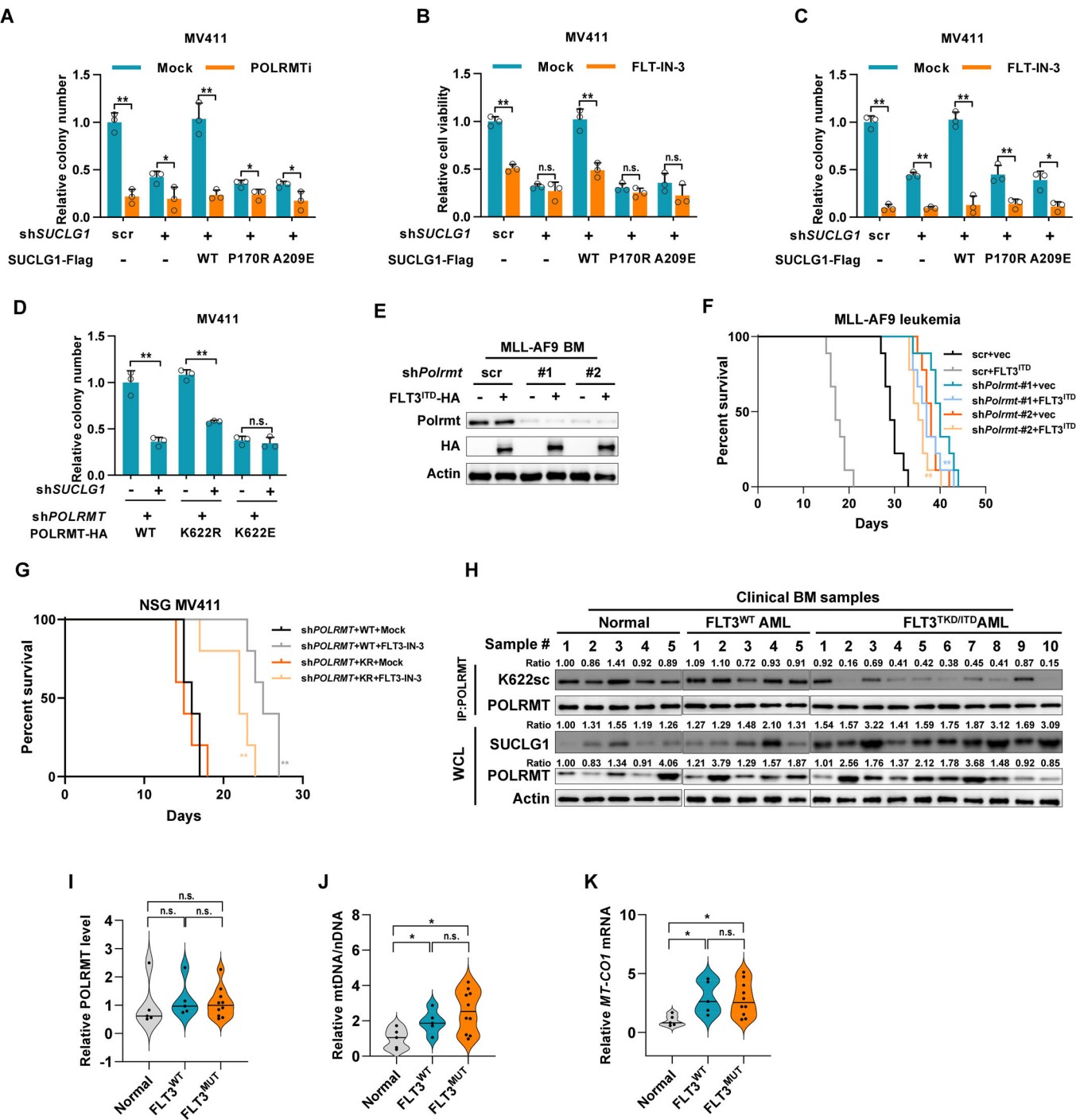

**Figure EV6. SUCLG1 suppresses POLRMT succinylation to support leukemic proliferation.**

(**A**) *SUCLG1*-knockdown and rescue MV411 cells were treated with POLRMT inhibitor (POLRMTi). Colony formation abilities were assayed. Data represent means ± SD, $n = 3$ independent biological replicates, $t$ test. **$P < 0.01$; *$P < 0.05$. (**B, C**) *SUCLG1*-knockdown and rescue MV411 cells were treated with FLT3 inhibitor FLT-IN-3. Cell proliferation (**B**) and colony formation (**C**) were assayed. Data represent means ± SD, $n = 3$ independent biological replicates, $t$ test. **$P < 0.01$; *$P < 0.05$; n.s. not significant. (**D**) Scrambled control or shRNA against *SUCLG1* (#1) were stably expressed in *POLRMT*-knockdown and rescue MV411 cells. Colony formation was determined. Data represent means ± SD, $n = 3$ independent biological replicates, $t$ test. **$P < 0.01$; n.s. not significant. (**E, F**) MLL-AF9 bone marrow cells were transduced with vector control or HA-tagged FLT3$^{ITD}$. Scrambled control or shRNAs targeting *Polrmt* was stably expressed. Protein expression was determined by western blotting (**E**). Animal survival was assayed after bone marrow transplantation (**F**). $n = 9$ mice, log-rank (Mantel–Cox) test. **$P < 0.01$. (**G**) *POLRMT*-knockdown and rescue MV411 cells were transplanted into sublethally irradiated NSG mice. Mice were treated with or without FLT3-IN-3, animal survival was determined. $n = 5$ mice, log-rank (Mantel–Cox) test. **$P < 0.01$. (**H–K**) Human clinical AML bone marrow samples were collected. K622 succinylation of immunopurified POLRMT was determined by western blotting (**H**). Protein expression of POLRMT (**I**), mtDNA abundance (**J**), and *MT-CO1* mRNA expression (**K**) were determined. $n = 5$–10, $t$ test. *$P < 0.05$; n.s. not significant. Source data are available online for this figure.

