## [Peer Review File · The EMBO Journal]

SUCLG1 restricts POLRMT succinylation to enhance mitochondrial biogenesis and leukemia progression

Wei-Wei Yan, Chengmei Xie, Sijun Sun, Quan Zheng, Jingyi Wang, Zihao Wang, Cheuk-Him Man, Haiyan Wang, Yunfan Yang, Tianshi Wang, Leilei Shi, Shengjie Zhang, Chen Huang, Shuangnian Xu, and Yi-Ping Wang

Corresponding authors: Yi-Ping Wang (yiping.wang1@shgh.cn), Shuangnian Xu (xushuangnian@tmmu.edu.cn), Shengjie Zhang (shengjie.zhang@shgh.cn), Chen Huang (chen.huang2@shgh.cn)

Review Timeline:

Submission Date:	27th Sep 23
Editorial Decision:	17th Nov 23
Revision Received:	14th Feb 24
Editorial Decision:	19th Mar 24
Revision Received:	21st Mar 24
Accepted:	22nd Mar 24

Editor: Daniel Klimmeck

Transaction Report:

Dear Dr Wang,

Thank you again for submitting your manuscript EMBOJ-2023-115734 for consideration by the EMBO Journal. Please accept my sincere apologies for getting back to you with unusual protraction due to delayed referee input, as well as detailed discussion in the editorial team. As indicated, your manuscript has been seen by three referees with expertise in leukemia / cancer biology and cellular metabolism, and we have received reports from all of them, which are shown below.

Given the referees' positive recommendations, I would like to invite you to submit a revised version of the manuscript, addressing the comments of all three reviewers. I should add that it is EMBO Journal policy to allow only a single round of revision, and acceptance of your manuscript will therefore depend on the completeness of your responses in this revised version.

I would appreciate if you could contact me during the next weeks for exchange e.g. a video call to discuss your perspective on the comments and potential plan for revisions.

Please feel free to contact me if you have any questions or need further input on the referee comments.

When submitting your revised manuscript, please carefully review the instructions below.

Please feel free to approach me any time should you have additional questions related to this.

Thank you for the opportunity to consider your work for publication.

I look forward to your revision.

Kind regards,

Daniel Klimmeck

Daniel Klimmeck, PhD
Senior Editor
The EMBO Journal

Instruction for the preparation of your revised manuscript:

2) individual production quality figure files as .eps, .tif, .jpg (one file per figure).

3) a .docx formatted letter INCLUDING the reviewers' reports and your detailed point-by-point response to their comments. As part of the EMBO Press transparent editorial process, the point-by-point response is part of the Review Process File (RPF), which will be published alongside your paper.

4) a complete author checklist, which you can download from our author guidelines ([https://wol-prod-cdn.literatumonline.com/pb-assets/embo-site/Author Checklist%20-%20EMBO%20J-1561436015657.xlsx](https://wol-prod-cdn.literatumonline.com/pb-assets/embo-site/Author%20Checklist%20-%20EMBO%20J-1561436015657.xlsx)). Please insert information in the checklist that is also reflected in the manuscript. The completed author checklist will also be part of the RPF.

6) It is mandatory to include a 'Data Availability' section after the Materials and Methods. Before submitting your revision, primary datasets produced in this study need to be deposited in an appropriate public database, and the accession numbers and database listed under 'Data Availability'. Please remember to provide a reviewer password if the datasets are not yet public (see <https://www.embopress.org/page/journal/14602075/authorguide#datadeposition>).

7) Our journal encourages inclusion of *data citations in the reference list* to directly cite datasets that were re-used and obtained from public databases. Data citations in the article text are distinct from normal bibliographical citations and should directly link to the database records from which the data can be accessed. In the main text, data citations are formatted as follows: "Data ref: Smith et al, 2001" or "Data ref: NCBI Sequence Read Archive PRJNA342805, 2017". In the Reference list, data citations must be labelled with "[DATASET]". A data reference must provide the database name, accession number/identifiers and a resolvable link to the landing page from which the data can be accessed at the end of the reference. Further instructions are available at .

8) At EMBO Press we ask authors to provide source data for the main and EV figures. Our source data coordinator will contact you to discuss which figure panels we would need source data for and will also provide you with helpful tips on how to upload and organize the files.

Numerical data can be provided as individual .xls or .csv files (including a tab describing the data). For 'blots' or microscopy, uncropped images should be submitted (using a zip archive or a single pdf per main figure if multiple images need to be supplied for one panel). Additional information on source data and instruction on how to label the files are available at .

9) We replaced Supplementary Information with Expanded View (EV) Figures and Tables that are collapsible/expandable online (see examples in <https://www.embopress.org/doi/10.15252/emboj.201695874>). A maximum of 5 EV Figures can be typeset. EV Figures should be cited as 'Figure EV1, Figure EV2' etc. in the text and their respective legends should be included in the main text after the legends of regular figures.

11) For data quantification: please specify the name of the statistical test used to generate error bars and P values, the number (n) of independent experiments (specify technical or biological replicates) underlying each data point and the test used to calculate p-values in each figure legend. The figure legends should contain a basic description of n, P and the test applied. Graphs must include a description of the bars and the error bars (s.d., s.e.m.).

We realize that it is difficult to revise to a specific deadline. In the interest of protecting the conceptual advance provided by the work, we recommend a revision within 3 months (15th Feb 2024). Please discuss the revision progress ahead of this time with

the editor if you require more time to complete the revisions.

Referee #1:

Yan and colleagues provide insights in how SUCLG1 regulates disease progression in mouse and humanized leukemia models. They revealed that SUCLG1 controls mitochondrial mass and mtDNA-encoded gene expression. They also showed that SUCLG1 restricts POLRMT succinylation by reducing mitochondrial succinyl-CoA levels known to regulate mitobiogenesis. Mechanistically, Yan and colleagues demonstrate that K622 is a major succinylation site of POLMRT that is repressed by SUCLG1 to maintain mitochondrial respiration and transcription. Next, they discover that SUCLG1 is upregulated in FLT3 mutants leukemia thereby repressing POLMRT succinylation to increase mitobiogenesis. Functionally, they displayed in leukemia development that SUCLG1 and POLRMT are important to support FLT3-driven cancer progression. They nicely showed that FLT3 mutant leukemia upregulate SUCLG1 to promote AML progression by enhancing mitobiogenesis.

Overall, this is a very well-written and presented manuscript describing an interesting and highly clinically relevant mechanism of mitochondrial metabolism regulating cancer progression. The authors present their findings clearly and describe them accurately within the text using mainly genetic approaches. A real strength of this study is the identification of a novel mechanism of AML progression and metabolic dependency. However, there are some open questions in the study design and the current set of data as listed below.

Below are comments that aim to further increase the clarity and significance of this exciting study:

1. Can the authors please provide a more detailed metabolic analysis of tumor cells with SUCLG1 knockdown including glucose uptake measurements, ROS levels measurements, proliferations/apoptosis measurements, senescence measurements and cell cycle measurements?
2. The authors should provide direct measurements of complex I,II, III, IV and V activities in SUCLG1 knockdown cells.
3. Can the authors provide some explanation of how cancer cells survive which such a dysregulation of mtDNA levels vs nDNA levels (Fig. 1D/E)?
4. In Fig. 2B the authors should provide the metabolic data for glycolysis, other TCA metabolites (e.g. citrate/ malate) and amino acids for a complete picture of metabolic changes upon SUCLG1 knockdown.
5. Do the authors detect an increase in glutamine uptake when they knockdown SUCLG1?
6. Do the authors detect differences in OGDH expression upon SUCLG1 knockdown?
7. Are SUCLG1 deficient cells auxotrophic for pyruvate and uridine? Can the functional defects upon SUCLG1 knockdown be rescued by supplementing the cell culture medium with uridine and pyruvate?
8. The authors should provide metabolic analysis including complex activity assays in SUCLG1 deficient leukemia cells in Figure 4.

Referee #2:

Manuscript Review for Yan, Weiwei et al. SUCLG1 restricts POLRMT succinylation to enhance mitochondrial biogenesis and leukemia progression. EMBO Journal

SUMMARY

Mitochondrial gene expression is tightly linked to metabolic state, but the mechanisms that link metabolite levels to transcription of mitochondrial DNA remain poorly described. In the manuscript "SUCLG1 restricts POLRMT succinylation to enhance mitochondrial biogenesis and leukemia progression", Yan et al. present compelling data supporting a model in which the activity of the TCA enzyme SUCLG1 modulates mitochondrial DNA expression by altering the affinity of the mitochondrial RNA polymerase POLRMT for mitochondrial DNA via post-translational succinylation. The work is impressive and the findings are novel and fit well within the scope and readership of EMBO Journal. Additionally, the authors provide clear clinical relevance to a

basic biological finding.

However, there are a few areas that warrant further exploration/modification.

Major points

1. POLRMT mediates mtDNA replication as well as mtDNA transcription. It is possible that the decrease in mitochondrial DNA content, mitochondrial number, and mtDNA transcription after SUCLG1 knockdown are due to diminished mtDNA replication, rather than impaired enzymatic activity of POLRMT. It is important to understand the impact of succinylated POLRMT on mtDNA replication dynamics, and uncouple these from its effects on transcriptional regulation. One way of assessing this, which would not require development of novel experimental methods or reagents, would be to utilize the in organello transcription assay. If the K622R POLRMT mutant rescues mitobiogenesis following SUCLG1 knockdown by increasing mtDNA transcription, we would expect a level of in organello transcription similar to WT. Conversely, the impairment in mitochondrial number, biomass and oxidative function may be due to failure of POLRMT to mediate mtDNA replication. The authors should include the K622R POLRMT mutants in Figures 3D-E and 3H to more rigorously implicate mtDNA transcription as the cause of SUCLG1-mediated mitobiogenesis.
2. The authors argue that the impaired mitochondrial function that results from SUCLG1 knock-down is a result of the accumulation of succinyl-CoA and consequent succinylation of POLRMT. However, the enzymatic function of SUCLG1 in the TCA cycle may be equally important and should be controlled for. One way of doing this would be to knockdown or inhibit OGDH, which should decrease the levels of Succinyl-CoA. If necessary, succinate levels and TCA function could be rescued by supplementation with cell-permeable succinate.
3. A remaining question in the implication of the model is how FLT3 regulates SUCLG1 expression. Figure 4 suggests that FLT3 controls a more general set of mitochondrial genes to upregulate mitochondrial activity. The authors should define this set of genes and at a minimum identify consensus sequences in enhancers/promoters for transcriptional regulators that would explain their joint regulation by the FLT3 pathway. In Figure 5D, the results suggest that FLT3 is doing more than simply regulating SUCLG1 levels to increase mitochondrial biomass since SUCLG1 expression cannot increase cell viability of FLT3-IN. Similarly, the relationship between SUCLG1 and POLRMT in the context of leukemia is not linear (Figure I); while POLRMT-KR is maximally lethal in a SUCLG1-dependent manner (orange curve has highest mortality and grey has lowest), SUCLG1 knockdown has intermediate survival in the POLRMT-KR mutant setting (yellow curve). This suggests additional functions of SUCLG1 which may be related to its enzymatic function (see major point 2) or alternative functions possibly related to succinylation. Lastly, in the discussion the authors state "... higher expression of SUCLG1 was linked to worse overall survival in FLT3WT patients, but not in FLT3-TKD/ITD individuals (Fig 5O)." This seems to contradict the hypothesis that higher SUCLG1 levels contribute to decreased succinylation and thus increased activity of POLRMT, driving leukemogenesis. In fact, it is only in the FLT3-WT setting that SUCLG1 high patients have worse survival than SUCLG1 low. Given these findings, the authors should revise these statements and allow for additional functions of FLT3 and SUCLG1 in regulating mitochondrial biogenesis and contributing to leukemia development.

Minor points

- There is a typo in the last sentence of the introduction: "induce POLRMT hypomethylation" should be "induce POLRMT hyposuccinylation"
- It would be preferable to see the raw data with replicates instead of heat maps in figures, where possible (e.g. Fig 1C, D, J).
- Fig 1A and E, does decreased SUCLG1 lead to altered expression of ETC correlated genes?
- In Fig 2B, why do succinate levels not decrease dramatically after SUCLG1 knockdown?
- The model implies cells with high rates of ATP or GTP synthesis should also have POLRMT hyposuccinylation and increased mitobiogenesis. Can the authors test this, perhaps by assessing POLRMT succinylation in highly translationally active cell lines? Or perhaps comment in the discussion.

The authors should expand the discussion to include the following points:

- Is regulation of POLRMT succinylation equally present in SUCLG2 and SUCLA2 expressing cells? (ie cell carrying out primarily anabolic versus catabolic metabolism)
- If succinylation is non-enzymatic, why would an association between POLRMT and SUCLG1 suggest that the latter is mediating succinylation of the former?
- SDH mutations are found in a variety of human malignancies. Does SDH loss or inhibition impair mitochondrial biogenesis through product inhibition of SUCLG1 by succinate and thus higher succinylation of POLRMT?

Referee #3:

Mitochondrial homeostasis crosstalks with cellular metabolism, it is critical for cellular physiology and can go awry in tumorigenesis.

The authors identified that the levels of SUCLG1, which converts succinyl-CoA to succinate, are positively associated with electron transport chain gene expression across various cancers. Succinylation of the mitochondrial polymerase (POLRMT)

negatively controls its activity and is critical for mitochondrial homeostasis. The manuscript showcases that high SUCLG1 levels and activity in cancer, reduces succinyl-coA levels and concomitantly POLRMT succinylation in cancer, leading to increased POLRMG1 activity and levels of electron transport chain transcripts. This is critical for mitochondrial biogenesis and cancer growth.

The manuscript is important for the field, and the studies are generally supported by data. I have a few suggestions that can help clarify aspects of the story, as it is outlined below:

-How does mutant FLT3 regulate SUCLG1 transcription in cancer? what are the critical transcription factors for this regulation? ideally, the authors should provide ChIP data for candidate transcription factors. can silencing of those factors rescue the effect caused by FLT3 hyperactivation?

-"Loss of SUCLG1 elevated POLRMT succinylation (suc-Lys), but not its lysine methylation (me-Lys), lysine acetylation (ac-Lys), or tyrosine phosphorylation (p-Tyr)": have the authors tested POLRMT ubiquitination levels?

-What are the levels and activity of the other TCA cycle enzymes, especially the ones downstream of SUCLG, such as SDHA/B etc, and SIRT5, that are encoded in the nuclear DNA, in mutant FLT3 background?

-How do the levels and succinylation of the mitochondrial DNA polymerase change in mutant FLT3 background?

-What is the effect of SIRT5 inhibition on POLRMT succinylation and ETC transcription in mutant FLT3 background?

-Are FLT3 mutant cells more sensitive to mitochondrial/ETC inhibitors, such as rotenone or antimycin?

Point-by-point response to Referees' comments

Referee #1:

Yan and colleagues provide insights in how SUCLG1 regulates disease progression in mouse and humanized leukemia models. They revealed that SUCLG1 controls mitochondrial mass and mtDNA-encoded gene expression. They also showed that SUCLG1 restricts POLRMT succinylation by reducing mitochondrial succinyl-CoA levels known to regulate mitobiogenesis. Mechanistically, Yan and colleagues demonstrate that K622 is a major succinylation site of POLMRT that is repressed by SUCLG1 to maintain mitochondrial respiration and transcription. Next, they discover that SUCLG1 is upregulated in FLT3 mutants leukemia thereby repressing POLMRT succinylation to increase mitobiogenesis. Functionally, they displayed in leukemia development that SUCLG1 and POLRMT are important to support FLT3-driven cancer progression. They nicely showed that FLT3 mutant leukemia upregulate SUCLG1 to promote AML progression by enhancing mitobiogenesis.

Response: Thank you very much for the positive comments.

Overall, this is a very well-written and presented manuscript describing an interesting and highly clinically relevant mechanism of mitochondrial metabolism regulating cancer progression. The authors present their findings clearly and describe them accurately within the text using mainly genetic approaches. A real strength of this study is the identification of a novel mechanism of AML progression and metabolic dependency. However, there are some open questions in the study design and the current set of data as listed below.

Response: Thank you for considering our work “interesting and highly clinically relevant”. We are also grateful for the open questions that inspired us to improve our work.

Below are comments that aim to further increase the clarity and significance of this exciting study:

1. Can the authors please provide a more detailed metabolic analysis of tumor cells with SUCLG1 knockdown including glucose uptake measurements, ROS levels measurements, proliferations/apoptosis measurements, senescence measurements and cell cycle measurements?

Response: Following your suggestion, we performed a more detailed metabolic analysis of *SUCLG1*-knockdown cells. Glucose consumption rates were increased in *SUCLG1*-knockdown cells, which is observed in both MV411 and HL60 stable cells (New Fig 2D and EV2F).

Further, CellROX Green staining of MV411 cells revealed increased ROS levels after depleting *SUCLG1* (new Fig EV1I-J). Bromodeoxyuridine (BrdU) incorporation was remarkably decreased in *SUCLG1*-knockdown cells, indicating reduced cell proliferation (new Fig EV1N). Increased apoptosis was also observed in *SUCLG1*-knockdown cells in Annexin V/7-amino-actinomycin (7-AAD) staining assay (new Fig EV1K-L).

Additionally, we treated MV411 cells with a low dose of doxorubicin to induce senescence. Loss of *SUCLG1* didn't result in obvious changes in β -galactosidase staining (new Fig EV1M). In agreement with the BrdU incorporation assay, cell cycle analysis demonstrated that *SUCLG1*-depleted cells were less actively proliferating compared to the scrambled control (new Fig EV1O).

2. The authors should provide direct measurements of complex I, II, III, IV and V activities in *SUCLG1* knockdown cells.

Response: As you suggested, we isolated mitochondria from stable cells (new Fig EV1E) and determined complexes activities using assay kits. Loss of *SUCLG1* resulted in decreased activities of Complexes I-V (new Fig 1H).

New Fig EV1E

New Fig 1H

Mitochondrial ETC complexes

3. Can the authors provide some explanation of how cancer cells survive which such a dysregulation of mtDNA levels vs nDNA levels (Fig. 1D/E)?

Response: To address your concern, we determined glycolytic fluxes and cellular ATP levels. Metabolic flux assay showed that *SUCLG1*-knockdown cells had increased basal glycolytic rate (new Fig 2C and EV2E), indicating that glycolysis may be enhanced to compensate for mitochondria respiration defects caused by impaired mitochondrial gene expression. Supporting this idea, lactate production was elevated after *SUCLG1* depletion in both MV411 and HL60 cells (new Fig 2E and EV2G).

New Fig 2C

New Fig EV2E

New Fig 2E

New Fig EV2G

Quantification of cellular ATP further validated that loss of *SUCLG1* resulted in decreased energy supply in MV411 and HL60 cells (new Fig 2F and EV2H). These results indicate that, in the state of *SUCLG1* depletion, glycolysis is upregulated to cope with mitochondria dysfunction and maintain cell survival.

4. In Fig. 2B the authors should provide the metabolic data for glycolysis, other TCA metabolites (e.g. citrate/ malate) and amino acids for a complete picture of metabolic changes upon *SUCLG1* knockdown.

Response: Following your suggestion, we performed metabolome profiling of *SUCLG1*-knockdown cells. Glycolytic intermediates were increased in *SUCLG1*-knockdown MV411 cells (new Fig 2B). Amino acids were also dysregulated after depleting *SUCLG1*. Strikingly, succinyl-CoA was accumulated in *SUCLG1*-deficient cells. While citrate didn't show consistent changes after depleting *SUCLG1*, malate levels were decreased in *SUCLG1*-knockdown MV411 cells (new Fig 2B). These data would provide a more comprehensive picture of the metabolic landscape under *SUCLG1*-deficient conditions.

5. Do the authors detect an increase in glutamine uptake when they knockdown SUCLG1?

Response: Thank you for your suggestion. Interestingly, we observed a decrease in glutamine consumption in *SUCLG1*-knockdown MV411 cells (new Fig 2G). This data indicates that glutamine metabolism is potentially defective in *SUCLG1*-depleted cells.

New Fig 2G

6. Do the authors detect differences in OGDH expression upon SUCLG1 knockdown?

Response: Following your advice, we determined OGDH protein expression with western blotting experiments. No obvious differences in OGDH expression were observed upon *SUCLG1* knockdown (new Fig EV2D).

New Fig EV2D

7. Are SUCLG1 deficient cells auxotrophic for pyruvate and uridine? Can the functional defects upon SUCLG1 knockdown be rescued by supplementing the cell culture medium with uridine and pyruvate?

Response: Thank you very much for the insightful advice. As you suggested, we added uridine and pyruvate to the culture of control and *SUCLG1*-knockdown stable cells. Uridine, but not pyruvate, partially rescued the defect of mitochondrial respiration of *SUCLG1*-depleted cells (new Fig 2P-Q). The addition of both pyruvate and uridine didn't further increase oxygen consumption (new Fig 2R).

New Fig 2P

New Fig 2Q

New Fig 2R

We further assayed cell proliferation and colony formation in the presence of pyruvate and uridine. In line with the oxygen consumption results, uridine, but not pyruvate, partially restored the proliferation of *SUCLG1*-depleted cells (new Fig 2S-V and EV2O). Combined treatment with both pyruvate and uridine didn't provide a further growth advantage to MV411 cells (new Fig 2V and EV2O). These observations suggest that *SUCLG1*-deficient cells are auxotrophic for uridine.

8. The authors should provide metabolic analysis including complex activity assays in *SUCLG1* deficient leukemia cells in Figure 4.

Response: As you suggested, we directly measured the activities of ETC complexes in FLT3-overexpressing CD34⁺ CB cells. FLT3^{ITD} remarkably increased ETC complexes activities (new Fig 4K). Depletion of *SUCLG1* reduced ETC complexes activities to similar levels in both control and FLT3^{ITD}-expressing cord blood cells (new Fig 4K), suggesting that mutant FLT3 relies on *SUCLG1* to upregulate mitochondrial respiration.

Further, we tested ETC complexes activities in *SUCLG1*-knockdown MV411 cells. While silencing either *FLT3* or *SUCLG1* downregulated complexes activities, co-depletion of both genes didn't result in a further decrease in complexes activities (new Fig 4R). Interestingly, *FLT3* signaling showed specificities towards different respiration complexes as complex IV activity was not significantly altered in *FLT3*-knockdown cells (new Fig 4R). Nevertheless, these data demonstrate that *FLT3* and *SUCLG1* cooperatively maintain ETC activity.

Referee #2:

Manuscript Review for Yan, Weiwei et al. *SUCLG1* restricts POLRMT succinylation to enhance mitochondrial biogenesis and leukemia progression. *EMBO Journal*

SUMMARY

Mitochondrial gene expression is tightly linked to metabolic state, but the mechanisms that link metabolite levels to transcription of mitochondrial DNA remain poorly described. In the manuscript "SUCLG1 restricts POLRMT succinylation to enhance mitochondrial biogenesis and leukemia progression", Yan et al. present compelling data supporting a model in which the activity of the TCA enzyme *SUCLG1* modulates mitochondrial DNA expression by altering the affinity of the mitochondrial RNA polymerase POLRMT for mitochondrial DNA via post-translational succinylation. The work is impressive and the findings are novel and fit well within the scope and readership of *EMBO Journal*. Additionally, the authors provide clear clinical relevance to a basic biological finding.

Response: Thank you very much for considering our work compelling, impressive, and novel.

However, there are a few areas that warrant further exploration/modification.

Response: We are grateful for the elegant experiments designed by the referee which provide an opportunity for us to deeply explore mechanisms of *SUCLG1*-mediated mitochondrial biogenesis.

Major points

1. POLRMT mediates mtDNA replication as well as mtDNA transcription. It is possible that the decrease in mitochondrial DNA content, mitochondrial number, and mtDNA transcription after *SUCLG1* knockdown are due to diminished mtDNA replication, rather

than impaired enzymatic activity of POLRMT. It is important to understand the impact of succinylated POLRMT on mtDNA replication dynamics, and uncouple these from its effects on transcriptional regulation. One way of assessing this, which would not require development of novel experimental methods or reagents, would be to utilize the in organello transcription assay. If the K622R POLRMT mutant rescues mitobiogenesis following *SUCLG1* knockdown by increasing mtDNA transcription, we would expect a level of in organello transcription similar to WT. Conversely, the impairment in mitochondrial number, biomass and oxidative function may be due to failure of POLRMT to mediate mtDNA replication. The authors should include the K622R POLRMT mutants in Figures 3D-E and 3H to more rigorously implicate mtDNA transcription as the cause of *SUCLG1*-mediated mitobiogenesis.

Response: Thanks a lot for the insightful question and kindly designed experiments. Following your suggestions, we investigated the effect of K622R mutant on mitobiogenesis. *POLRMT*^{K622R} mutant rescues mitochondrial transcription to a level similar to wild-type *POLRMT* (new Fig 3D). Knockdown of *SUCLG1* in K622R-rescued cells had a minor effect on mtDNA transcription (new Fig 3E). In agreement, re-expression of K622R mutant restored oxygen consumption to a level comparable to wild-type *POLRMT* (new Fig 3H). These data demonstrate that *POLRMT*-mediated mtDNA transcription contributes to *SUCLG1*-mediated mitobiogenesis.

2. The authors argue that the impaired mitochondrial function that results from *SUCLG1* knock-down is a result of the accumulation of succinyl-CoA and consequent succinylation of *POLRMT*. However, the enzymatic function of *SUCLG1* in the TCA cycle may be equally important and should be controlled for. One way of doing this would be to knockdown or inhibit OGDH, which should decrease the levels of Succinyl-CoA. If necessary, succinate

levels and TCA function could be rescued by supplementation with cell-permeable succinate.

Response: Thank you a lot for the kind advice. Following your suggestion, we stably expressed shRNAs against *OGDH* in MV411 cells. Knockdown of *OGDH* reduced succinyl-CoA level and K622 succinylation in MV411 cells (new Fig 3I and EV3L). Treatment with cell-permeable succinate (dimethyl succinate, DMS) had a negligible effect on POLRMT succinylation (new Fig 3I and EV3M). Supplementation with DMS didn't show a significant impact on mtDNA-encoded gene expression, mitochondrial mass and respiration in control cells, but remarkably enhanced mitobiogenesis and respiration in *OGDH*-knockdown cells (new Fig 3J-L). These data suggest that both enzymatic activity of *SUCLG1* in TCA cycle and *SUCLG1*-mediated hyposuccinylation of POLRMT are necessary for mitochondrial function.

3. A remaining question in the implication of the model is how FLT3 regulates *SUCLG1* expression. Figure 4 suggests that FLT3 controls a more general set of mitochondrial genes to upregulate mitochondrial activity. The authors should define this set of genes and at a minimum identify consensus sequences in enhancers/promoters for transcriptional regulators that would explain their joint regulation by the FLT3 pathway.

Response: As you suggested, we set out to identify FLT3-regulated mitochondrial genes. The impact of FLT3 on leukemia transcriptome has been previously profiled in MOLM14 and MV411 cells (Sabatier et al. Cancer Discov, 2023, PMID: 37012202). We collected the differentially expressed mitochondrial genes (new Fig 5A-B) from the published dataset, among which 25 genes were strongly downregulated (Fold change ≥ 2 ; $p < 0.01$) in both cell

lines after depleting *FLT3* (new Fig 5C). We validated that these genes were downregulated in *FLT3*-knockdown MV411 cells (new Fig 5D). Analysis of potential promoter regions of these genes revealed at least three different consensus sequences (new Fig 5E-G and EV5A-C), which potentially allows transcription regulators to jointly regulate their expression in response to *FLT3* signaling.

New Fig 5A

New Fig 5B

New Fig 5C

New Fig 5D

New Fig 5E

New Fig 5F

New Fig 5G

New Fig EV5A

Consensus #1		
QDR	-443	CAAAGTCTGGGATTACA -426
YRDC	-302	CAAAGTCTGGGATTACA -285
DTYMK	-455	CAAAGTCTGGGATTACA -438
OAT	-351	TAAAGTCTGGGATTACA -334
MTHFD1L	-786	CAAAGTCTGGGATTACA -769
PGAM5	-903	CAAAGTCTGGGATTACA -886
UNG	-267	CAAAGTCTGGGATTACA -250
SFXN1	-297	CAAAGTCTGGGATTACA -280
HSPB1	-537	CAGAGTCTGGGATTACA -520
SLC25A19	-142	CTGAATGCTGGGATTACA -125
GPT2	-833	CTTGTGCTGGGATTACA -816
PRDX4	-750	AAAAGTCTGGGATTACA -733
FASN	-802	GAAAGTCTGGGATTACA -785
FXN	-372	GAAAGTCTGGGATTACA -355
MALSU1	-646	TCAAAGTCTGGGATTACA -629
HPDL	-946	CACAATCTGGGATTACA -929
TIMM10	-397	CTGACTGCTGGGATTACA -380
SLC25A10	-497	TGAGTAACTGGGATTACA -480
TIMM17A	-647	CGAGTGGGCGGATTACA -630
SPR	-586	GTTACTGCTGGGATTACA -569
RP1A	-361	CTCACTGCTGGGATTACA -344
MRPS12	-795	GAAAGTCTGGGATTACA -778
TOMM5	-66	GACCTCTGGGATTACA -49
BOLA3	-867	GAAACTGCTGGGATTACA -850
TOMM40	-286	CAAAGTCTGGGATTACA -269
Consensus		CAAAGTCTGGGATTACA

New Fig EV5B

Consensus #2		
OAT	-530	ACCTCCCTCTCTGG -516
QDR	-623	ACCTCCCTCTCTGG -609
MTHFD1L	-966	ACCTCCCTCTCTGG -952
SLC25A10	-541	ACCTCCCTCTCTGG -527
DTYMK	-596	ACCTCCCTCTCTGG -582
FXN	-342	GATCCCTCTCTGG -328
TIMM17A	-41	ACTCCCTCTCTGG -27
YRDC	-452	ACATCAGCTCTCTGG -438
HSPB1	-177	TCTCCCTCTCTGG -163
MALSU1	-492	ACCTCCCTCTCTGG -478
RP1A	-562	AGCTCCCTCTCTGG -548
SPR	-30	CGAACCCTCTCTGG -16
GPT2	-916	ACCTCCCTCTCTGG -902
HPDL	-46	CCCTCTCTCTCTGG -32
MRPS12	-449	AGCTCCCTCTCTGG -435
PGAM5	-104	TGCCCTCTCTCTGG -90
PRDX4	-157	CGTCCCTCTCTGG -143
SFXN1	-87	CCCTCCCTCTCTGG -73
SLC25A19	-972	ACATCTGCTCTCTGG -958
TOMM40	-333	AGGCTCTCTCTCTGG -319
UNG	-117	ACCTCCCTCTCTGG -103
FASN	-56	CCACCTCTCTCTGG -42
TIMM10	-602	AGCTCCCTCTCTGG -588
TOMM5	-684	ACATCAGCTCTCTGG -670
BOLA3	-340	ACCGAGCTCTCTGG -326
Consensus		ACCTCCCTCTCTGG

New Fig EV5C

Consensus #3		
FXN	-653	GAAACCCGGAGGC -641
TIMM10	-702	GAAACCCGGAGGC -690
TIMM17A	-493	GAAACCCGGAGGC -481
UNG	-652	GAAACCCGGAGGC -640
HPDL	-510	GAAACCCGGAGGC -498
HSPB1	-757	GAAACCCGGAGGC -745
FASN	-832	GAAACCCGGAGGC -820
MRPS12	-261	GAAACCCGGAGGC -249
TOMM40	-248	GAAACCCGGAGGC -236
SFXN1	-104	GAAACCCGGAGGC -92
GPT2	-71	GAAACCCGGAGGC -59
PRDX4	-37	GAAACCCGGAGGC -25
SLC25A10	-847	GAAACCCGGAGGC -835
SPR	-277	GAAACCCGGAGGC -265
BOLA3	-345	GAAACCCGGAGGC -333
DTYMK	-738	GAAACCCGGAGGC -726
MALSU1	-17	GAAACCCGGAGGC -5
MTHFD1L	-85	GAAACCCGGAGGC -73
PGAM5	-87	GAAACCCGGAGGC -75
SLC25A19	-301	GAAACCCGGAGGC -289
QDR	-839	GAAACCCGGAGGC -827
OAT	-914	GAAACCCGGAGGC -902
RP1A	-74	GAAACCCGGAGGC -62
TOMM5	-700	GAAACCCGGAGGC -688
YRDC	-711	GAAACCCGGAGGC -699
Consensus		GAACCCGGAGGC

In Figure 5D, the results suggest that FLT3 is doing more than simply regulating SUCLG1 levels to increase mitochondrial biomass since SUCLG1 expression cannot increase cell viability of FLT3-IN. Similarly, the relationship between SUCLG1 and POLRMT in the context of leukemia is not linear (Figure I); while POLRMT-KR is maximally lethal in a SUCLG1-dependent manner (orange curve has highest mortality and grey has lowest), SUCLG1 knockdown has intermediate survival in the POLRMT-KR mutant setting (yellow curve). This suggests additional functions of SUCLG1 which may be related to its enzymatic function (see major point 2) or alternative functions possibly related to succinylation.

Response: We totally agree with you that the relationship between SUCLG1 and POLRMT is not linear. To examine the contribution of POLRMT succinylation to cell proliferation, we depleted *SUCLG1* in *POLRMT*-knockdown and rescue cells. Loss of *SUCLG1* also demonstrated a growth-suppressive effect in cells re-expressing *POLRMT*^{K622R} (new Fig 6C-D and EV6D), suggesting that SUCLG1 may have functions other than modulating POLRMT succinylation to support leukemia cell proliferation. We have revised the corresponding statements to clarify the relationship between FLT3 and SUCLG1 in the context of AML.

Lastly, in the discussion the authors state "... higher expression of SUCLG1 was linked to worse overall survival in FLT3WT patients, but not in FLT3-TKD/ITD individuals (Fig 5O)." This seems to contradict the hypothesis that higher SUCLG1 levels contribute to decreased succinylation and thus increased activity of POLRMT, driving leukemogenesis. In fact, it is only in the FLT3-WT setting that SUCLG1 high patients have worse survival than SUCLG1 low. Given these findings, the authors should revise these statements and allow for additional functions of FLT3 and SUCLG1 in regulating mitochondrial biogenesis and contributing to leukemia development.

Response: We are grateful to you for pointing out this issue. Our original hypothesis was that *SUCLG1* overexpression in the context of FLT3-mutant leukemia provides sufficient growth advantage compared to FLT3-WT leukemia. In this situation, excess expression of *SUCLG1* is not able to provide further advantage to FLT3-mutated leukemia. To clarify this issue, we compared the mRNA expression of *SUCLG1* in different groups of TCGA AML. Unexpectedly, *FLT3*^{TKD/ITD}*SUCLG1*^{low} group didn't show higher *SUCLG1* expression compared to *FLT3*^{WT} leukemia (Figure for referee #2). This observation supports your idea that SUCLG1 and FLT3 have additional functions in regulating leukemia development. We thank the reviewer for helping us gain more insights into the biological function of SUCLG1

and revised the overstatements accordingly.

Figure for referee #2

Minor points

- There is a typo in the last sentence of the introduction: "induce POLRMT hypomethylation" should be "induce POLRMT hyposuccinylation"

Response: We sincerely apologize for the hastiness when preparing the manuscript. The typo has been corrected in the revised version.

- It would be preferable to see the raw data with replicates instead of heat maps in figures, where possible (e.g. Fig 1C, D, J).

Response: As you suggested, we have included the raw data with replicates in the revised version.

- Fig 1A and E, does decreased SUCLG1 lead to altered expression of ETC correlated genes?

Response: Following your question, we performed a correlation analysis of *SUCLG1* and ETC-correlated genes. *SUCLG1* showed a positive correlation with other ETC-correlated metabolic genes in the majority of TCGA cancer types (new Fig EV1F). We further tested the mRNA expression of ETC-correlated genes in *SUCLG1*-knockdown cells. Two of the top-ranking ETC-related genes, *PRDX5* and *TXN2*, showed decreased mRNA levels in both MV411 and HL60 cells (new Fig EV1G-H). These data indicate that ETC-correlated genes may work cooperatively to control mitochondrial activity.

- In Fig 2B, why do succinate levels not decrease dramatically after *SUCLG1* knockdown?
 Response: To address your question, we tested the mRNA expression of genes related to succinate metabolism and transport in mitochondria, cytosol, and cell membrane (new Fig EV2I). *SLC13A5*, a cell membrane-bound transporter of succinate (Zwart et al. J Pharmacol Exp Ther. 2015, PMID: 26324167), showed increased expression after knocking down *SUCLG1* (new Fig 2I). Other succinate transporters only demonstrated modest changes in their mRNA expression. *SLC13A5* may import exogenous succinate in response to decreased succinate production in *SUCLG1*-knockdown cells, leading to a less pronounced decrease of succinate.

New Fig EV2I

New Fig 2I

- The model implies cells with high rates of ATP or GTP synthesis should also have POLRMT hyposuccinylation and increased mitobiogenesis. Can the authors test this, perhaps by assessing POLRMT succinylation in highly translationally active cell lines? Or perhaps comment in the discussion.

Response: Thank you for the enlightening question. Following your advice, we aimed to determine the correlation between POLRMT succinylation and mitochondrial translation. MRPL45 is a key factor in mitochondrial ribosome that determines protein translational activity (Greber et al. Nature. 2014, PMID: 24362565). We performed correlation analysis in transcriptomes of the cancer cell line encyclopedia (Ghandi et al. Nature. 2019, PMID: 31068700). Strikingly, mRNA expression of *SUCLG1* showed a significantly positive correlation with *MRPL45* (new Fig 3M). This data indicates that *SUCLG1* activity is potentially higher in translationally active cell lines. We adopted *MRPL45*^{low} (HS578T and HCC1806) and *MRPL45*^{high} (KM12 and A549) cell lines for further analysis. Nucleotide quantification revealed that *MRPL45*^{high} cell lines had higher levels of ATP and GTP (new Fig 3N), which were accompanied by decreased POLRMT K622 succinylation (new Fig

30). The link between POLRMT succinylation and energy production potentially couples mitochondrial transcription and translation to ensure efficient production of mtDNA-encoded proteins.

The authors should expand the discussion to include the following points:

- Is regulation of POLRMT succinylation equally present in SUCLG2 and SUCLA2 expressing cells? (ie cell carrying out primarily anabolic versus catabolic metabolism)

Response: Thank you for the insightful suggestion. We have added the following discussion in the revised version: "SUCLG2 and SUCLA2 primarily carry out anabolic and catabolic functions, respectively (Gut et al., 2020). Inherent genetic mutations of succinyl-CoA ligase predominantly targeted SUCLA2 and SUCLG1 (Hadrava Vanova et al., 2022). Disease-causing mutations of SUCLG2 are rarely reported and their role in mtDNA transcription remains obscure (Carrozzo et al, 2016). This intriguing phenomenon may be explained in different ways: (1) POLRMT succinylation is intimately linked to succinate-CoA-mediated ATP production, which is majorly present in highly energy-consuming organs where catabolism primarily occurs; (2) it is possible that SUCLA2 could compensate for the loss of SUCLG2, but not vice versa. Further efforts are required to elucidate the functional diversifications of succinate-CoA ligase subunits in mitochondria metabolism".

- If succinylation is non-enzymatic, why would an association between POLRMT and SUCLG1 suggest that the latter is mediating succinylation of the former?

Response: Recent studies of high-resolution imaging of small molecules revealed that metabolites are not homogeneously distributed within a cellular compartment (Li et al. Cell Metab, 2023, PMID: 36309010). Previous studies drove us to identify SUCLG1 binding partners to investigate its biological function. For example, OGDH enzyme which controls the level of α -KG was reported to enter the nucleus of plant cells. Importantly, OGDH restricted α -KG level and directly interacted with α -KG-dependent histone demethylases to control locus-specific histone demethylation (Huang et al. Science, 2023, PMID: 37440635). Another example comes from UDP-glucose 6-dehydrogenase (UGDH). UGDH is a key enzyme in the uronic acid pathway and directly interacts with Hu antigen R (HuR),

an RNA-binding protein that binds to and stabilizes many short-lived mRNAs. UDP-glucose suppressed the mRNA binding of HuR to promote mRNA decay. UGDH bound to HuR and decreased UDP-glucose to specifically stabilize SNAI1 mRNA, contributing to tumor metastasis (Wang et al. Nature, 2019, PMID: 31243371). In our work, SUCLG1 may more efficiently modulate the succinylation of target proteins by directly interacting with them. In the revised text, we discussed the potential advantage of the physical association between POLRMT and SUCLG1 in regulating mitochondrial biogenesis.

- SDH mutations are found in a variety of human malignancies. Does SDH loss or inhibition impair mitochondrial biogenesis through product inhibition of SUCLG1 by succinate and thus higher succinylation of POLRMT?

Response: We thank the reviewer for the inspiring comment. To test this possibility, we stably depleted *SDHA* and observed a significant accumulation of succinate and succinyl-CoA in MV411 cells, accompanied by increased K622 succinylation of endogenous POLRMT (new Fig EV3H-I). In line with these findings, both mtDNA copy number and MTG intensity were decreased in *SDHA*-knockdown cells (new Fig EV3J-K).

Referee #3:

Mitochondrial homeostasis crosstalks with cellular metabolism, it is critical for cellular physiology and can go awry in tumorigenesis.

The authors identified that the levels of SUCLG1, which converts succinyl-CoA to succinate, are positively associated with electron transport chain gene expression across various cancers. Succinylation of the mitochondrial polymerase (POLRMT) negatively controls its activity and is critical for mitochondrial homeostasis. The manuscript showcases that high

SUCLG1 levels and activity in cancer, reduces succinyl-coA levels and concomitantly POLRMT succinylation in cancer, leading to increased POLRMG1 activity and levels of electron transport chain transcripts. This is critical for mitochondrial biogenesis and cancer growth.

The manuscript is important for the field, and the studies are generally supported by data. I have a few suggestions that can help clarify aspects of the story, as it is outlined below:

Response: Thank you a lot for considering our study important and for helping us to improve the clarity.

-How does mutant FLT3 regulate SUCLG1 transcription in cancer? what are the critical transcription factors for this regulation? ideally, the authors should provide ChIP data for candidate transcription factors. can silencing of those factors rescue the effect caused by FLT3 hyperactivation?

Response: To address your concern, we reviewed the literature and collected a list of 16 transcription factors (TFs) that are known to be regulated by FLT3 and set out to define the transcriptional regulation of *SUCLG1* by FLT3.

FLT3-targeted TFs list for referee #3				
Transcription factor	Full name	Regulation by FLT3	Journal	PMID
ASH2L	Set1/Ash2 histone methyltransferase complex subunit ASH2	Activation	Leukemia and Lymphoma, 2017	28185526
ATF4	Activating Transcription Factor 4	Activation	Cancer Discovery, 2021	33436370
BCL6	B-cell lymphoma 6 protein homolog	Suppression	Blood Advances, 2021	34614509
CEBPA	CCAAT enhancer-binding protein alpha	Suppression	Blood, 2012	22474248
CTNNB1	Catenin beta-1	Activation	European Journal of Haematology, 2012	22126602
E2F1	E2F transcription factor 1	Activation	Biochemical Pharmacology, 2023	36803956
FOXO1	Forkhead box protein O1	Suppression	Blood, 2020	32315388
FOXO3	Forkhead box protein O3	Suppression	Blood, 2020	32315388
GLI2	Zinc finger protein GLI2	Activation	Science Translational Medicine, 2015	26062848
MYC	MYC proto-oncogene	Activation	Journal of Biological Chemistry, 2004	15067010
NFATC1	Nuclear factor of activated T-cells, cytoplasmic 1	Activation	Leukemia, 2015	25976987
RUNX1	Runt-related transcription factor 1	Activation	Cell Reports, 2015	26212328
SIRT1	NAD-dependent protein deacetylase sirtuin-1	Activation	FASEB Journal, 2022	36440960
SOX6	Transcription factor SOX-6	Suppression	Journal of Cancer Research and Clinical Oncology, 2023	36117190
STAT5A	Signal transducer and activator of transcription 5A	Activation	Blood, 2007	17356133
TFEB	Transcription factor EB	Activation	FASEB Journal, 2022	36440960

We established a shRNA library against the known transcription regulators (three different shRNA for each TF) and transduced them into MV411 cells. Strikingly, depleting *E2F1* strongly suppressed *SUCLG1* expression, which was observed with three different shRNAs (new Fig 5H). Similar effects were observed in *E2F1*-knockdown MOLM14 cells (new Fig EV5D-E), further supporting *E2F1* as a potential governor of *SUCLG1* transcription.

New Fig EV5D

New Fig EV5E

SUCLG1 protein was downregulated in *E2F1*-depleted MV411 cells, which is coupled with hypersuccinylation of POLRMT (new Fig 5I). Treatment of *E2F1*-knockdown cell with FLT3 inhibitor didn't further elevate POLRMT succinylation (new Fig 5I).

New Fig 5I

We further examined *E2F1* binding with the *SUCLG1* gene. Genomic DNA sequence around the transcription start site (TSS) of *SUCLG1* contains at least two different *E2F1*-binding motifs (new Fig 5J). We performed ChIP-qPCR assay with an *E2F1*-specific antibody. The occupancy of *E2F1* at *SUCLG1* gene was readily detected in both MV411 and MOLM14 cells (new Fig 5K-L). *E2F1* binding of *SUCLG1* was strongly weakened in *E2F1*-knockdown cells, supporting the specificity of the ChIP antibody (new Fig 5K-L).

More importantly, *E2F1* depletion suppressed the cellular respiration of MV411. Treatment with FLT3 inhibitor didn't further reduce oxygen consumption in *E2F1*-knockdown cells (new Fig 5M). Collectively, *E2F1* potentially mediates FLT3 signaling to enhance *SUCLG1* expression.

New Fig 5J

New Fig 5K

New Fig 5L

New Fig 5M

"Loss of *SUCLG1* elevated POLRMT succinylation (suc-Lys), but not its lysine methylation (me-Lys), lysine acetylation (ac-Lys), or tyrosine phosphorylation (p-Tyr)": have the authors tested POLRMT ubiquitination levels?

Response: Following your advice, we constructed a Myc-tagged POLRMT expression vector. POLRMT-Myc and HA-tagged ubiquitin were co-expressed in MV411 cells. POLRMT protein was immunoprecipitated with Myc antibody. Detection of HA-ubiquitin displayed no obvious changes of immunopurified POLRMT in control or *SUCLG1*-knockdown cells (new Fig EV2L). Besides, we immunopurified endogenous POLRMT and its ubiquitination signal didn't show remarkable changes (new Fig EV2M). These data suggest that *SUCLG1* had no effect on POLRMT ubiquitination.

New Fig EV2L

New Fig EV2M

-What are the levels and activity of the other TCA cycle enzymes, especially the ones downstream of SUCLG, such as SDHA/B etc, and SIRT5, that are encoded in the nuclear DNA, in mutant FLT3 background?

Response: As you suggested, we tested protein expression in various leukemia cell lines. Notably, SUCLG1, but not SDHA, SDHB, FH, SIRT5 or POLG, was overexpressed in FLT3-mutated cell lines (MONOMAC6, MOLM14, and MV411) (new Fig 4A).

In addition, we analyzed POLRMT-related genes expression in TCGAAML. FH and POLG, but not the other tested genes, showed enhanced mRNA expression in FLT3-mutated samples (new Fig EV4N). Interestingly, treatment of MV411 cells with FLT3 inhibitors (FLT3-IN-3 and AC220) had a negligible effect on protein expression of SDH, FH, SIRT5, and POLG (new Fig EV4O).

We also assayed the effect of FLT3 on the enzymatic activities of SDH and FH. SDH, but not FH, showed increased activities in CD34⁺ CB cells expressing mutant FLT3 (new Fig EV4B-C). Besides, treatment of MV411 cells with FLT3 inhibitors decreased the catalytic activity of SDH, but not FH (new Fig EV4P-Q). These results suggest that SDH, but not FH, is a target for mutant FLT3. The expression and activity may be differentially regulated by SUCLG1 and FLT3 signaling in leukemia cells.

-How do the levels and succinylation of the mitochondrial DNA polymerase change in mutant FLT3 background?

Response: To address your question, we determined POLG protein expression in different leukemia cells. POLG didn't show obvious changes in FLT3-WT or mutated cell lines (new Fig 4A). We further overexpressed FLT3^{ITD} into CD34⁺ cord blood cells. Protein expression of POLG was minimally altered (new Fig EV4G). We also treated MV411 cells with FLT3 inhibitor and immunopurified endogenous POLG, of which lysine succinylation showed negligible changes (new Fig EV4R).

-What is the effect of SIRT5 inhibition on POLRMT succinylation and ETC transcription in mutant FLT3 background?

Response: Following your advice, we treated control or FLT3^{ITD}-expressing CD34⁺ CB cells

with a chemical inhibitor against SIRT5. SIRT5 inhibition upregulated POLRMT succinylation and reduced mtDNA binding of POLRMT, which is accompanied by decreased expression of mtDNA-encoded genes (*MT-CO1*, *MT-ND6*, and *MT-ATP6*) (new Fig EV4D-F). In *FLT3^{ITD}*-expressing cells, SIRT5 inhibition altered POLRMT succinylation and mtDNA binding to a lesser extent (new Fig EV4D-E). A similar pattern was observed in the mRNA expression of mtDNA-encoded genes (new Fig EV4F).

-Are *FLT3* mutant cells more sensitive to mitochondrial/ETC inhibitors, such as rotenone or antimycin?

Response: Thank you for providing this angle to help us rethink the work. To address your question, we treated various leukemia cell lines with increasing doses of rotenone and monitored cell survival. As anticipated, cell lines carrying *FLT3* mutations (MV411, MOLM14, and MONOMAC6) displayed increased sensitivities to rotenone (new Fig 6E). We also performed a cell survival assay of control and *FLT3^{ITD}*-expressing CD34⁺ cord blood cells. Overexpressing *FLT3^{ITD}* sensitized CD34⁺ cord blood cells to rotenone (new Fig 6F). These results suggest that *FLT3* mutation confers a dependency on oxidative phosphorylation and renders cells more vulnerable to ETC inhibition.

Dear Dr Yi-Ping Wang,

Thank you for submitting your revised manuscript (EMBOJ-2023-115734R) to The EMBO Journal. Your amended study was sent back to the three referees for their scientific reevaluation, and we have received detailed comments from all of them, which I enclose below. As you will see, the experts state that the work has been substantially improved by the revisions and they are now broadly in favour of publication.

Thus, we are pleased to inform you that your manuscript has been accepted in principle for publication in The EMBO Journal.

We now need you to take care of a number of issues related to formatting and data presentation as detailed below, which should be addressed at re-submission.

Please contact me at any time if you have additional questions related to below points.

As you might have seen on our web page, every paper at the EMBO Journal now includes a 'Synopsis', displayed on the html and freely accessible to all readers. The synopsis includes a 'model' figure as well as 2-5 one-short-sentence bullet points that summarize the article. I would appreciate if you could provide this figure and the bullet points.

Thank you for giving us the chance to consider your manuscript for The EMBO Journal. I look forward to your final revision.

Again, please contact me at any time if you need any help or have further questions.

Best regards,

Daniel Klimmeck

>> Authors: All corresponding authors need institutional email addresses entered into our online system and accounts need the ORCID number linked (S.X., C.H.). Please see below for additional information.

>> Author Contributions: Please remove the author contributions information from the manuscript text. Note that CRediT has replaced the traditional author contributions section as of now because it offers a systematic machine-readable author contributions format that allows for more effective research assessment. and use the free text boxes beneath each contributing author's name to add specific details on the author's contribution.

>> Appendix: To make it easier to read, the appendix table should rather be renamed Dataset EV1 and uploaded as a dataset file in excel format.

>> Please cite referencing of your previous work (PMID: 33770508) in the Material and Methods section.

>> Consider additional changes and comments from our production team as indicated below:

- Figure legends:

1. Please indicate the statistical test used for data analysis in the legends of figures 5a-c; EV 1b.
2. Please note that information related to n is missing in the legend of figure EV 4n.
3. Please note that the error bar is not defined in the legend of figure 1h.
4. Please note that the red arrowheads are not defined in the legend of figure 1g. This needs to be rectified.

Please note that as of January 2016, our new EMBO Press policy asks for corresponding authors to link to their ORCID iDs. You can read about the change under "Authorship Guidelines" in the Guide to Authors here: <http://emboj.embopress.org/authorguide>

In order to link your ORCID iD to your account in our manuscript tracking system, please do the following:

1. Click the 'Modify Profile' link at the bottom of your homepage in our system.
2. On the next page you will see a box half-way down the page titled ORCID*. Below this box is red text reading 'To Register/Link to ORCID, click here'. Please follow that link: you will be taken to ORCID where you can log in to your account (or create an account if you don't have one)
3. You will then be asked to authorise Wiley to access your ORCID information. Once you have approved the linking, you will be brought back to our manuscript system.

We regret that we cannot do this linking on your behalf for security reasons. We also cannot add your ORCID iD number manually to our system because there is no way for us to authenticate this iD number with ORCID.

Thank you very much in advance.

Referee #1:

This is a revised version of the previously submitted manuscript from Yan and colleagues. The authors have performed a thorough revision of the manuscript and included extended series of novel observations and data.

The authors have fully addressed all of my comments, and therefore I recommend this work to be accepted for publication.

Referee #2:

The authors have adequately addressed my major and minor points. The manuscript provides sufficient evidence to support a model whereby succinylation of POLRMT impairs mitochondrial gene expression and biogenesis. This inhibitory effect is removed by conversion of succinyl-CoA to succinate by the enzyme SUCLG1.

I would recommend the authors emphasize that some of the phenotypes observed with SUCLG1 loss are also related to the loss of its canonical function, especially given the results from Figure 3 showing a rescue with DMS.

The manuscript in its current form is of general significance for those interested in mitochondrial metabolism, signaling and cancer metabolism.

Referee #3:

The authors have adequately addressed my questions and comments. I think the study is of high impact.

Point-by-point response to referees and editor's comments

Referee #1:

This is a revised version of the previously submitted manuscript from Yan and colleagues. The authors have performed a thorough revision of the manuscript and included extended series of novel observations and data.

The authors have fully addressed all of my comments, and therefore I recommend this work to be accepted for publication.

Response: Thank you for your support and helping us improve our work.

Referee #2:

The authors have adequately addressed my major and minor points. The manuscript provides sufficient evidence to support a model whereby succinylation of POLRMT impairs mitochondrial gene expression and biogenesis. This inhibitory effect is removed by conversion of succinyl-CoA to succinate by the enzyme SUCLG1.

I would recommend the authors emphasize that some of the phenotypes observed with SUCLG1 loss are also related to the loss of its canonical function, especially given the results from Figure 3 showing a rescue with DMS.

The manuscript in its current form is of general significance for those interested in mitochondrial metabolism, signaling and cancer metabolism.

Response: We have modified the text to speak about the canonical function of SUCLG1. Thank you for supporting the significance of our work.

Referee #3:

The authors have adequately addressed my questions and comments. I think the study is of high impact.

Response: Thank you for evaluating our manuscript and considering our study as high impact.

Dear Dr Yi-Ping Wang,

Thank you for submitting the revised version of your manuscript. I have now evaluated your amended manuscript and concluded that the remaining minor concerns have been sufficiently addressed.

I am pleased to inform you that your manuscript has been accepted for publication in the EMBO Journal.

On a different note, I would like to alert you that EMBO Press offers a format for a video-synopsis of work published with us, which essentially is a short, author-generated film explaining the core findings in hand drawings, and, as we believe, can be very useful to increase visibility of the work. Please see the following link for representative examples and their integration into the article web page:

<https://www.embopress.org/doi/full/10.15252/emboj.2019103932>

Best regards,

Daniel Klimmeck

Daniel Klimmeck, PhD
Senior Editor
The EMBO Journal
EMBO
Postfach 1022-40
Meyerhofstrasse 1
D-69117 Heidelberg
contact@embojournal.org
Submit at: <http://emboj.msubmit.net>
